# EZHIP constrains Polycomb Repressive Complex 2 activity in germ cells

Roberta Ragazzini[1,8], Raquel Pérez-Palacios[1,8], Irem H. Baymaz[2], Seynabou Diop[1], Katia Ancelin [1], Dina Zielinski[1,3], Audrey Michaud[1], Maëlle Givelet[4], Mate Borsos [5,6], Setareh Aflaki[1], Patricia Legoix[7], Pascal W.T.C. Jansen[2], Nicolas Servant [1,3], Maria-Elena Torres-Padilla[5,6], Deborah Bourc'his[1], Pierre Fouchet[4], Michiel Vermeulen[2] & Raphaël Margueron[1]

The Polycomb group of proteins is required for the proper orchestration of gene expression due to its role in maintaining transcriptional silencing. It is composed of several chromatin modifying complexes, including Polycomb Repressive Complex 2 (PRC2), which deposits H3K27me2/3. Here, we report the identification of a cofactor of PRC2, EZHIP (EZH1/2 Inhibitory Protein), expressed predominantly in the gonads. EZHIP limits the enzymatic activity of PRC2 and lessens the interaction between the core complex and its accessory subunits, but does not interfere with PRC2 recruitment to chromatin. Deletion of *Ezhip* in mice leads to a global increase in H3K27me2/3 deposition both during spermatogenesis and at late stages of oocyte maturation. This does not affect the initial number of follicles but is associated with a reduction of follicles in aging. Our results suggest that mature oocytes *Ezhip*−/− might not be fully functional and indicate that fertility is strongly impaired in *Ezhip*−/− females. Altogether, our study uncovers EZHIP as a regulator of chromatin landscape in gametes.

[1] Institut Curie, PSL Research University, Sorbonne University, INSERM U934/ CNRS UMR3215, 26, rue d'Ulm, 75005 Paris, France. [2] Department of Molecular Biology, Faculty of Science, Radboud Institute for Molecular Life Sciences, Oncode Institute, Radboud University Nijmegen, Route 274 M850.03.87 Geert Grooteplein 28, Nijmegen 6525 GA, Netherlands. [3] INSERM U900, Mines ParisTech, 26, rue d'Ulm, Paris, France. [4] Commissariat à l'énergie atomique et aux énergies alternatives, Institut de radiobiologie cellulaire et moléculaire, Laboratoire des Cellules Souches Germinales, INSERM U967, Fontenay-aux-Roses 92260, France. [5] Institute of Epigenetics and Stem Cells (IES), Helmholtz Zentrum München, D-81377 München, Germany. [6] Faculty of Biology, Ludwig Maximilians Universität, D-81377 München, Germany. [7] ICGex Next-Generation Sequencing Platform, Curie Institute, 26, rue d'Ulm, Paris, France. [8] These authors contributed equally: Roberta Ragazzini, Raquel Pérez-Palacios. Correspondence and requests for materials should be addressed to R.M. (email: raphael.margueron@curie.fr)

Early in development, cells commit to specific lineages and acquire precise identities that require maintenance throughout the lifespan of the organism. Polycomb group proteins play an important role in this process by maintaining transcriptional repression through the regulation of chromatin structure[1]. In mammals, this machinery is composed of two main complexes: Polycomb Repressive Complex 1 and 2 (PRC1 and 2). The core PRC2 complex is composed of four subunits: the catalytic subunit EZH1/2, SUZ12, EED, and RbAp46/48[1]. PRC2 catalyzes the di- and tri-methylation of lysine 27 on histone H3 (H3K27me2/3), an enzymatic activity which is required for its function. Indeed, the mutation of lysine 27 of histone H3 to arginine leads to loss of gene repression, and mutant flies display a phenotype similar to deletion of PRC2 components[2]. H3K27me3 is generally enriched around the promoter of transcriptionally silent genes and contributes to the recruitment of PRC1[1]. H3K27me2 is widely distributed, covering 50–70% of histones, and its role is less defined but may be to prevent aberrant enhancer activation[3].

The question of how PRC2 is targeted to chromatin and how its enzymatic activity is controlled has received ongoing attention[4]. Cumulative evidence suggests that PRC2 may not be actively recruited to chromatin and that, instead its activity, is promoted by the recognition of its own mark H3K27me3, ubiquitination of lysine 119 of H2A, GC-richness, or by condensed chromatin[4]. Conversely, some histone modifications negatively influence PRC2 function, particularly those associated with active transcription, such as H3K4me3 and H3K36me3[4]. PRC2 binding to chromatin may also be inhibited by DNA methylation[5], although other reports suggest that PRC2 is compatible with DNA methylation[6].

A number of accessory subunits have now been shown to influence PRC2 function[4]. Recent comprehensive proteomic analyses suggest that they might form around two main PRC2 subtypes, PRC2.1 and PRC2.2[7]. The subunit SUZ12 plays a central role by orchestrating the cofactor interactions[8]. PRC2.1 includes one of the three Polycomb-like proteins (PHF1, MTF2 or PHF19) together with the recently identified PRC2 partners EPOP and PALI1[9,10]. The three Polycomb-like proteins harbor one Tudor domain and two PHD finger domains each[4]. Their Tudor domain is able to recognize H3K36me3 decorated genes, which could be important for PRC2 association with transcribed targets[4]. The function of EPOP remains ambiguous since, in vitro, it stimulates PRC2 catalytic activity while, in vivo, it limits PRC2 binding, likely through interaction with Elongin BC[11]. In contrast, PALI1 is required for H3K27me3 deposition both in vitro and in vivo[10]. The other complex, PRC2.2, includes JARID2 and AEBP2 subunits in equal stoichiometry[12,13]. Both are able to stimulate PRC2 catalytic activity in vitro with JARID2 being also able to bind nucleosomes[14]. JARID2 also appears to be necessary for PRC2 targeting at its loci, possibly through its DNA-binding domain or as a result of its methylation by PRC2[4]. AEBP2 binds to DNA in vitro, but appears to negatively modulate PRC2 in vivo[15,16]. Of note, AEBP2 was reported to stimulate PRC2 through a mechanism independent of PRC2 allosteric activation[17,18]. While we now have a good picture of the accessory subunits interacting with PRC2, their precise roles are only partially understood. This might be due to compensatory mechanisms, such that interfering both with PRC2.1 and PRC2.2 is required to inhibit PRC2 recruitment[19] as observed upon loss of SUZ12[20].

The regulation of chromatin structure in germ cells is pivotal, as these cells are the bridge between generations and therefore potential vector of epigenetic information. In particular, H3K27me3 has been shown to be involved in parental imprinting[21–23]. Yet, in contrast to the extensive characterization of PRC2 in models, such as mouse embryonic stem cells (ESC), much less is known about the regulation of its enzymatic activity in germ cells. Deletion of PRC2 core components during spermatogenesis results in the progressive loss of germ cells, indicating that its activity is required for this process[24,25]. At later stages of spermatogenesis, when round spermatids differentiate into mature sperm, histones are progressively replaced by protamines. A variable fraction of the genome retains a nucleosomal structure (1% in mice, 10–15% in human), with histones carrying posttranslational modifications, including H3K27me3[26]. During oogenesis, histones are maintained and H3K27me3 is detected throughout this process[27–29]. However, H3K27me3 displays a peculiar pattern of enrichment in the growing oocyte, showing broad enrichment in intergenic regions and gene deserts (reviewed in refs. [30,31]). Genetic interference with PRC2 function in growing oocytes does not prevent their maturation, but has been linked to a postnatal overgrowth phenotype in the progeny[32] possibly through the control of imprinting[23].

Here, we report the identification of a tissue-specific cofactor of PRC2, EZHIP. In human and mouse, we show that this cofactor is expressed primarily in gonads and that it limits PRC2-mediated H3K27me3 deposition. Inactivation of this cofactor in mice results in excessive deposition of this mark both during spermatogenesis and oogenesis. We further provide evidences that mutant oocytes with this altered epigenetic content are not fully functional, and that mouse female fertility is impaired.

## Results

**Identification of a cofactor of PRC2 in the gonad.** PRC2 recruitment and enzymatic activity is controlled by a set of cofactors interacting in a partially mutually exclusive manner with the core subunit SUZ12, but little is known about its regulation in germ cells. To tackle this question, we first focused on testes (more abundant material than ovaries), and took advantage of knock-in mouse models expressing an N-terminal Flag-tagged version of either EZH1 or EZH2 from their respective endogenous locus (this study and[33]). We verified the expression of the tagged-EZH1 by western blot on mouse testis nuclear extract, and were able to detect the presence of a slowly migrating polypeptide, which is specifically pulled down by Flag-Immunoprecipitation (Flag-IP) (Supplementary Fig. 1A). We then isolated nuclei from adult mouse testes (WT control, EZH2-Flag or EZH1-Flag), performed Flag-IP, and subjected the samples to mass spectrometry. The results of three independent IPs are represented as volcano plots (Fig. 1a; Supplementary Fig. 1B). As expected, both EZH1 and EZH2 proteins interact with the other PRC2 core components and with known accessory subunits: AEBP2, JARID2, PHF1, and MTF2 (Fig. 1a; Supplementary Fig. 1B). Interestingly, our experiments also reveal the existence of an additional partner, the uncharacterized protein AU022751 (ENSMUST00000117544; NM_001166433.1), which we retrieved in both EZH1 and EZH2 pulldowns. We referred to this cofactor as "EZHIP" for EZH1/2 inhibitory protein. Of note, this protein was previously identified in PRC2 interactomes of mouse embryonic stem cells, but its function was not further investigated[13,34,35]. In order to confirm this interaction, we overexpressed Flag-tagged versions of the mouse and human homologs in HeLa-S3 cells (Supplementary Fig. 1C), and performed IP followed by mass spectrometry. These reverse IPs confirmed the interaction between PRC2 and EZHIP (Fig. 1b; Supplementary Fig. 1D). Additional putative partners were identified in both IPs, but with the exception of USP7, they were not common to both homologs. We therefore did not pursue their study further. Importantly, these reverse IPs also indicate that EZHIP interacts with both PRC2 complex subtypes.

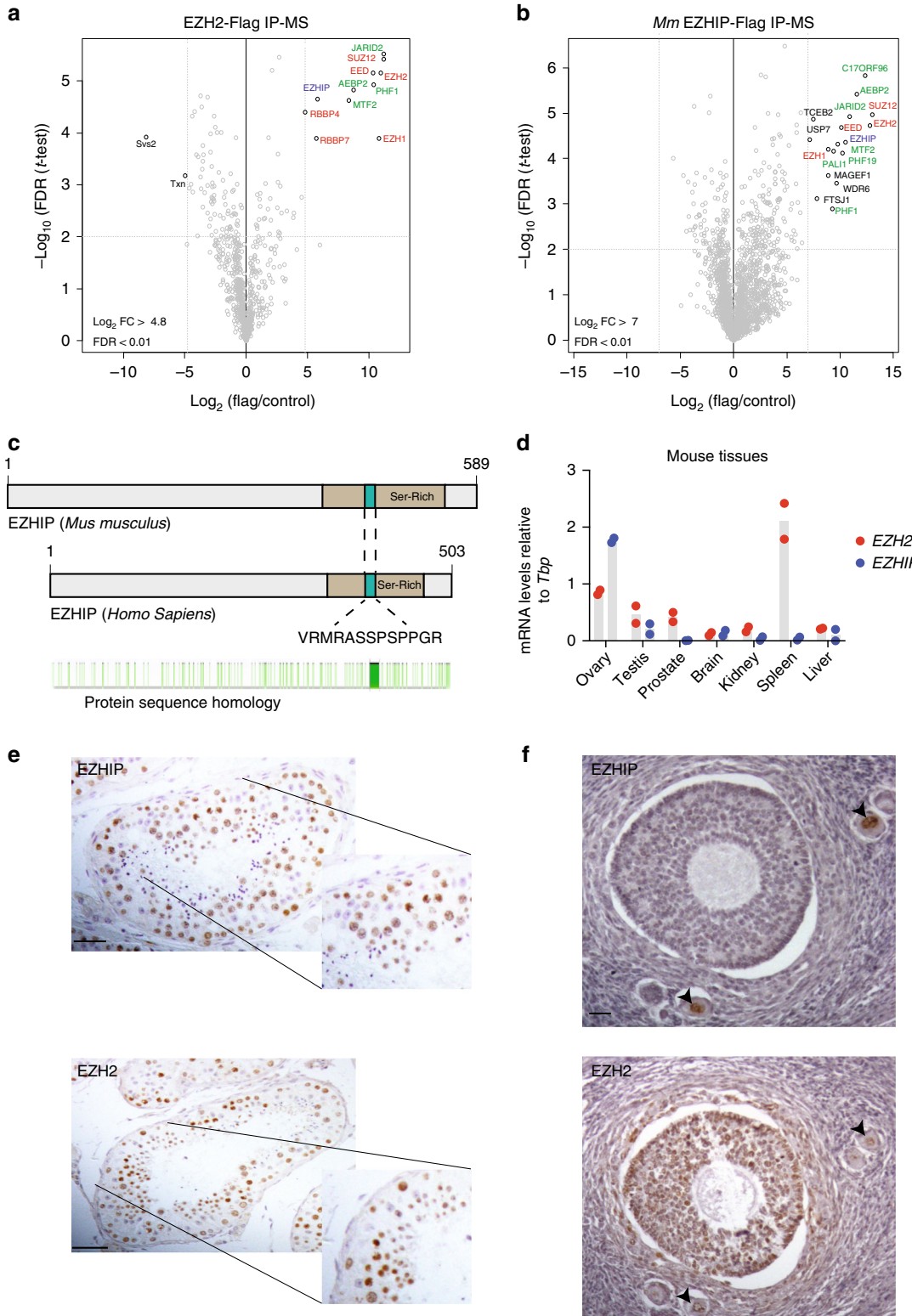

**Fig. 1** EZHIP interacts with PRC2 in gonads. **a** Volcano plot of EZH2 interactome from EZH2-Flag mice testis IP compared with WT. Core complex subunits are in red, in green the cofactors and in blue EZHIP, $n = 3$. **b** Volcano plot representation of EZHIP interactome after Flag-IP from HeLa-S3 overexpressing EZHIP compare with WT. Same color codes as in (**a**), $n = 3$. **c** Schematic representation of EZHIP protein sequence from *Mus Musculus* (upper part) and *Homo Sapiens* (middle part). Serine-rich region is colored in beige, and conserved amino acid stretch in green. The conserved sequence stretch is displayed as well as protein residues conservation between the two sequences in green (Sequence Homology determined using Genious software). **d** *Ezhip* and *Ezh2* mRNA relative abundance normalized to *Tbp* in various mice tissues (mean, $n = 2$). **e** EZHIP and EZH2 IHC staining on human adult seminiferous tubules sections. Representative result, $n \geq 5$. Scale bars, 50 μm. **f** EZHIP and EZH2 IHC staining on human adult ovaries sections, black arrows indicate the follicles. Representative result, $n \geq 5$. Scale bars, 30 μm

*EZHIP* is located on the X chromosome. In most species, it is a monoexonic gene—that may indicate that it was generated by retroposition—but in the mouse, splicing also creates a shorter isoform. Using phylogenetic analysis by maximum likelihood (PAML), we observed that *EZHIP* homologs are present across *Eutheria*, but we did not identify any homologs outside of this clade based on either sequence conservation or on synteny. *EZHIP* genes have rapidly evolved both at the nucleotide and amino acid levels, the rodent homologs being particularly distant from the rest (Fig. 1c; Supplementary Fig. 1E). This contrasts with the other PRC2 components, such as *EZH2*, which are highly conserved across mammals (Supplementary Fig. 1E). No known protein domain was predicted for EZHIP, and the only distinguishing feature is a serine-rich region (Fig. 1c in green), including a short amino acid stretch that is fully conserved in all orthologs identified (Fig. 1c in purple). To characterize *Ezhip* expression, we performed RT-qPCR on various tissues (3-month-old females and males). *Ezhip* mRNA expression was particularly high in ovaries; it was also expressed in testes, and much less in other tissues (Fig. 1d). Of note, *Ezhip* transcript level appears at least tenfold higher than any PRC2 core components or cofactors in oocytes (Supplementary Fig. 1F). *Ezhip*'s pattern of expression is distinct from that of *Ezh2*, which is expressed tissue wide, with the strongest expression observed in the spleen. Analysis of public gene expression data sets from fetal gonads[36] indicates *Ezhip* is preferentially expressed in E13.5 primordial germ cells (PGCs) compared with somatic cells, correlating with germ cell markers, such as *Piwil2* or *Prdm14* (Supplementary Fig. 1G). Interestingly, *Ezhip* belongs to a set of genes referred to as "germline-reprogramming-responsive" that become active following PGC DNA demethylation[37], as they are associated with strong CpG island promoters. Similarly, in humans *EZHIP* is highly transcribed in male and female PGCs from week 5 until week 9 of pregnancy, while almost absent in ESCs and somatic cells (Supplementary Fig. 1H)[38]. We confirmed this observation at the protein level by performing immunohistochemistry on sections of testes and ovaries of human origin. hEZHIP protein was detected in male germ cells inside the seminiferous tubules, especially in spermatogonia and round spermatids (Fig. 1e). In ovaries, EZHIP antibody stained primordial follicles and oocytes (red arrows), but not the external follicle cells in contrast to EZH2 antibody, which stained both zones (Fig. 1f). To summarize, EZHIP is a genuine cofactor of PRC2 in placental mammals. It is a fast-evolving protein with no known protein domain, it is expressed primarily in PGCs during development and remains present in the adult gonad.

**EZHIP is a negative regulator of PRC2 activity.** To study the molecular role of EZHIP, we sought a model cell line that would express this factor endogenously. The *EZHIP* transcript is undetectable from most cell lines, with the exception of U2OS, an osteosarcoma-derived cell line (Supplementary Fig. 2A). We used genome editing to generate U2OS clonal cells that were knockout for *EZHIP* or for *EED* as a control for PRC2 inactivation (U2OS *EZHIP*−/− and U2OS *EED*−/−, respectively). Both cell lines were viable, and had not overt phenotype. Western blot showed that deletion of *EED* destabilized the other PRC2 core components, such as EZH2, while inactivation of *EZHIP* had no discernible effect on the accumulation of these proteins (Fig. 2a). We then assessed H3K27 methylation and observed a robust increase in H3K27me2/3 upon *EZHIP* deletion, while H3K27me1 was stable and H3K27ac slightly reduced (Fig. 2b). Interestingly, H3K27me3 level was very low in U2OS compared with extract prepared from HEK-293T cells, which do not express *EZHIP* (Supplementary Fig. 2B). To confirm that *EZHIP* deletion was

directly responsible for the increased H3K27me3 in U2OS, we stably restored its expression using either full-length (FL) or deletion mutants (Fig. 2c) as verified by western blot (WB) and RT-q-PCR (Supplementary Fig. 2C). Upon re-expression of FL and mutant EZHIP, H3K27me3 returned to basal levels (Fig. 2c, d) with the notable exception of mutant M5 that lacks the conserved amino acids stretch (Fig. 2c). Given that such deletion abolishes EZHIP interaction with PRC2 in co-IP (mutant 6 vs. mutant 7; Supplementary Fig. 2D), it stands to reason that EZHIP likely regulates H3K27me3 deposition through direct interference with PRC2 activity.

To determine whether these alterations of PRC2 activity translate into aberrant gene expression, we analyzed the transcriptome of U2OS in the different genetic contexts described above by RNA-seq (Fig. 2d). Only a few genes were differentially expressed in U2OS *EED*−/− as compared with WT (FDR < 0.05), whereas ~500 genes were differentially expressed in *EZHIP*−/− vs. WT. The majority of which were downregulated, as expected considering the global gain of H3K27me3 repressive mark. Of note, gene ontology analysis of the genes downregulated upon EZHIP knockout did not reveal any robust categories (Supplementary Fig. 2E). Altogether, these results reveal that EZHIP inhibits the activity of PRC2 thus altering gene expression profile.

**Interplay between H3K27me2 & me3 upon expression of EZHIP.** Having shown the effects of EZHIP on PRC2 activity at the global level, we then investigated how this affects the chromatin landscape locally. First, we analyzed H3K27me3 genomic distribution in the absence of *EZHIP* by chromatin immuno-precipitation followed by sequencing (ChIP-seq). We used the U2OS *EED*−/−, in which H3K27me3 is not detectable as a negative control and compared it to the U2OS wild-type WT and *EZHIP*−/−. Replicates were well correlated and the U2OS WT and U2OS *EED*−/− clustered together, away from the U2OS *EZHIP*−/− (Supplementary Fig. 3A). This agrees with our earlier observation that H3K27me3 is very low in U2OS, as in the *EED* knockout (Supplementary Fig. 2B). In contrast, there was a genome-wide increase in H3K27me3 deposition upon deletion of *EZHIP* (Fig. 3a), as demonstrated by the large number of peaks detected in this context (Supplementary Fig. 3B).

To further characterize the role of EZHIP, we analyzed the genome-wide distribution of H3K27me2, H3K27ac, and H2Aub in U2OS WT vs. U2OS *EZHIP*−/− by "CUT&RUN"[39]. Replicates clustered together as expected (Supplementary Fig. 3C), and the correlation matrix revealed that H3K27me2 in the U2OS WT clusters with H3K27me3 in the *EZHIP*−/− condition, suggesting that EZHIP limits the conversion of H3K27me2 into H3K27me3.

This global correlation is confirmed when zooming on a specific region as illustrated by genome-browser screen shot (e.g. DRGX, Fig. 3b). At this locus, H2Aub and H3K27ac displayed only modest variations, and their enrichment appeared not particularly affected by the deletion of *EZHIP*. Focusing specifically on peaks that gain H3K27me3 upon deletion of *EZHIP* (Fig. 3c), we noticed a general decrease of H3K27me2 with a slight depletion around the peaks of H3K27me3 when comparing U2OS WT to U2OS *EZHIP*−/− (Fig. 3d). Of note, what happen for H3K27me2 at the regions gaining H3K27me3 in the U2OS *EZHIP*−/− does not reflect the genome-wide trend. Indeed, overall, we observe an increase of H3K27me2 illustrated by the increase number of peaks in the absence of EZHIP (Supplementary Fig. 3D). H3K27ac, which is known to antic-orrelate with H3K27me2/3 enrichment, was low and appeared to slightly decrease in the absence of EZHIP. Regarding H2Aub, the enrichment of the mark is only modestly increased by the deletion of *EZHIP* (Fig. 3d). If we focus on the TSS of genes either

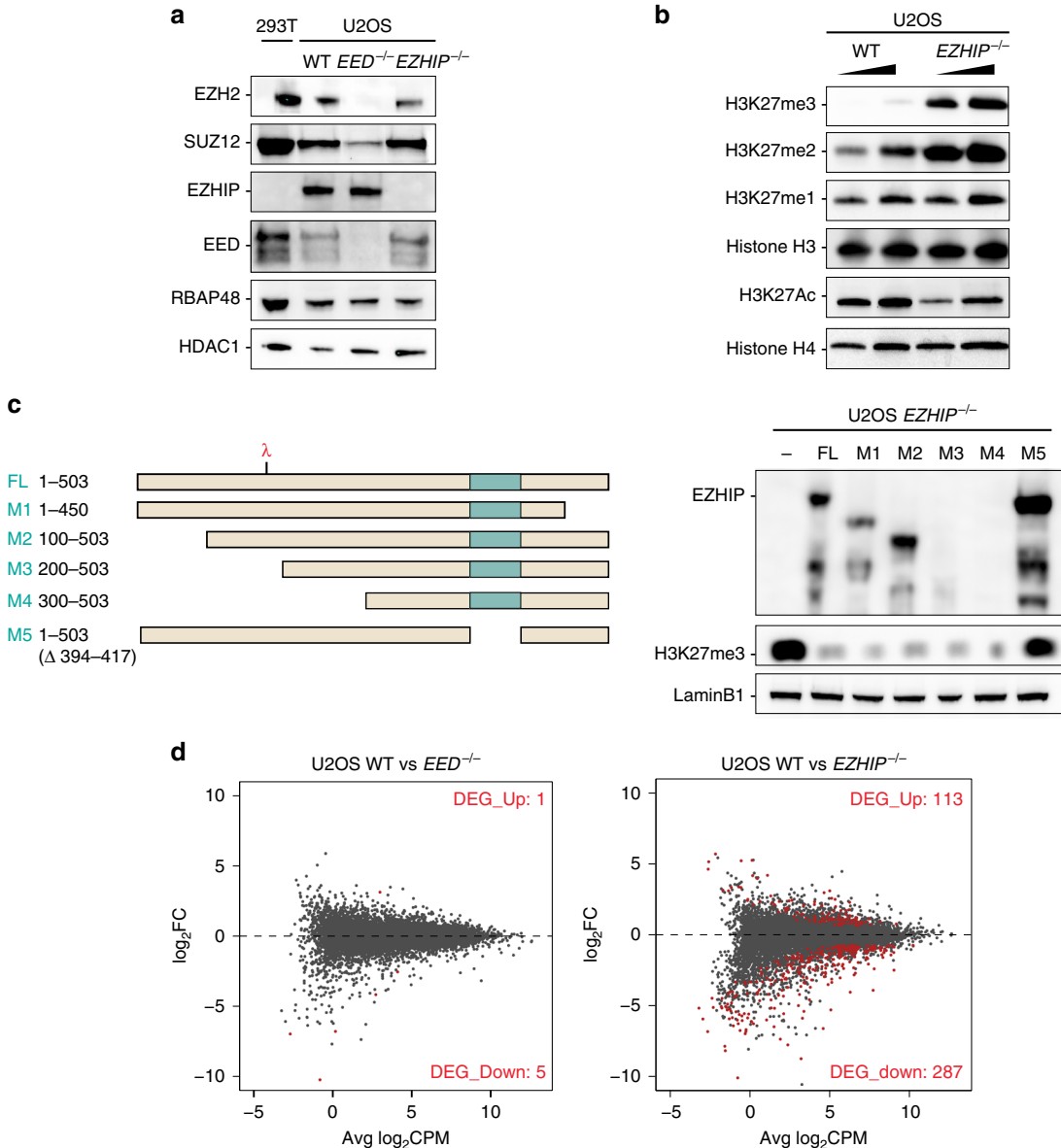

**Fig. 2** EZHIP inhibits H3K27me3 deposition. **a** Western blot analysis of PRC2 core complex subunits (SUZ12, RBAP48, EED, and EZH2), EZHIP and HDAC1 (loading control) on U2OS nuclear extracts WT, *EED−/−*, and *EZHIP−/−*. **b** Western blot analysis of H3K27 methylation: H3K27me1, H3K27me2, H3K27me3, H3K27ac and, H3 and H4 (loading controls). **c** Scheme representing EZHIP mutants (left panel) stably reintroduced in U2OS *EZHIP−/−* line. The red lambda indicates the epitope recognized by antibody detecting EZHIP protein. Western blot analysis (right panel) of cell lines expressing EZHIP FL or mutants probed with antibodies indicated on the left. **d** Mean-difference plot showing average log2 counts per million (logCPM) vs. log2 fold-change (logFC) expression between U2OS WT and *EED−/−*[69] or U2OS WT and *EZHIP−/−* (right). Right corner: the number of genes significantly differentially expressed (FDR < 0.05) corresponding to the dots colored in red. $n = 2$

upregulated or downregulated in the absence of EZHIP, we observed again a robust increase of H3K27me3 and an increase of H3K27ac for the downregulated genes and upregulated, respectively, whereas the other histone marks are only slightly affected (Fig. 3e). We conclude that EZHIP limits the activity of PRC2 favoring the deposition of H3K27me2 at regions normally enriched for H3K27me3. Of note, this altered chromatin landscape is reminiscent of what has been described upon expression of H3K27M mutant oncogenic histone[40].

**EZHIP impairs PRC2 activity, but not its binding to chromatin.** Considering the inhibitory action of EZHIP on PRC2, we next sought to explore the underlying mechanisms. First, we hypothesized that EZHIP could limit PRC2 binding to chromatin.

To test this hypothesis, we performed CUT&RUN against SUZ12 to monitor PRC2 recruitment to chromatin, comparing U2OS WT, U2OS EED−/−, and U2OS *EZHIP−/−*. Focusing again on DRGX, we observed that SUZ12 enrichment is lost in the absence of EED, whereas SUZ12 is enriched both in U2OS WT and U2OS *EZHIP−/−* (Fig. 4a). This result held true when we analyzed all the peaks that gain H3K27me3 in the absence of EZHIP (Supplementary Fig. 4A). Overall, we observed a slight increase of SUZ12 enrichment in particular at the regions flanking the peaks. Since our attempts to immunoprecipitate EZHIP were unsuccessful, we used immunofluorescence (IF) to evaluate its colocalization with EED and H3K27me2. The specificity of EZHIP antibody by IF is demonstrated by the lack of signal in U2OS EZHIP−/− (Supplementary Fig. 4B). In U2OS WT, EZHIP

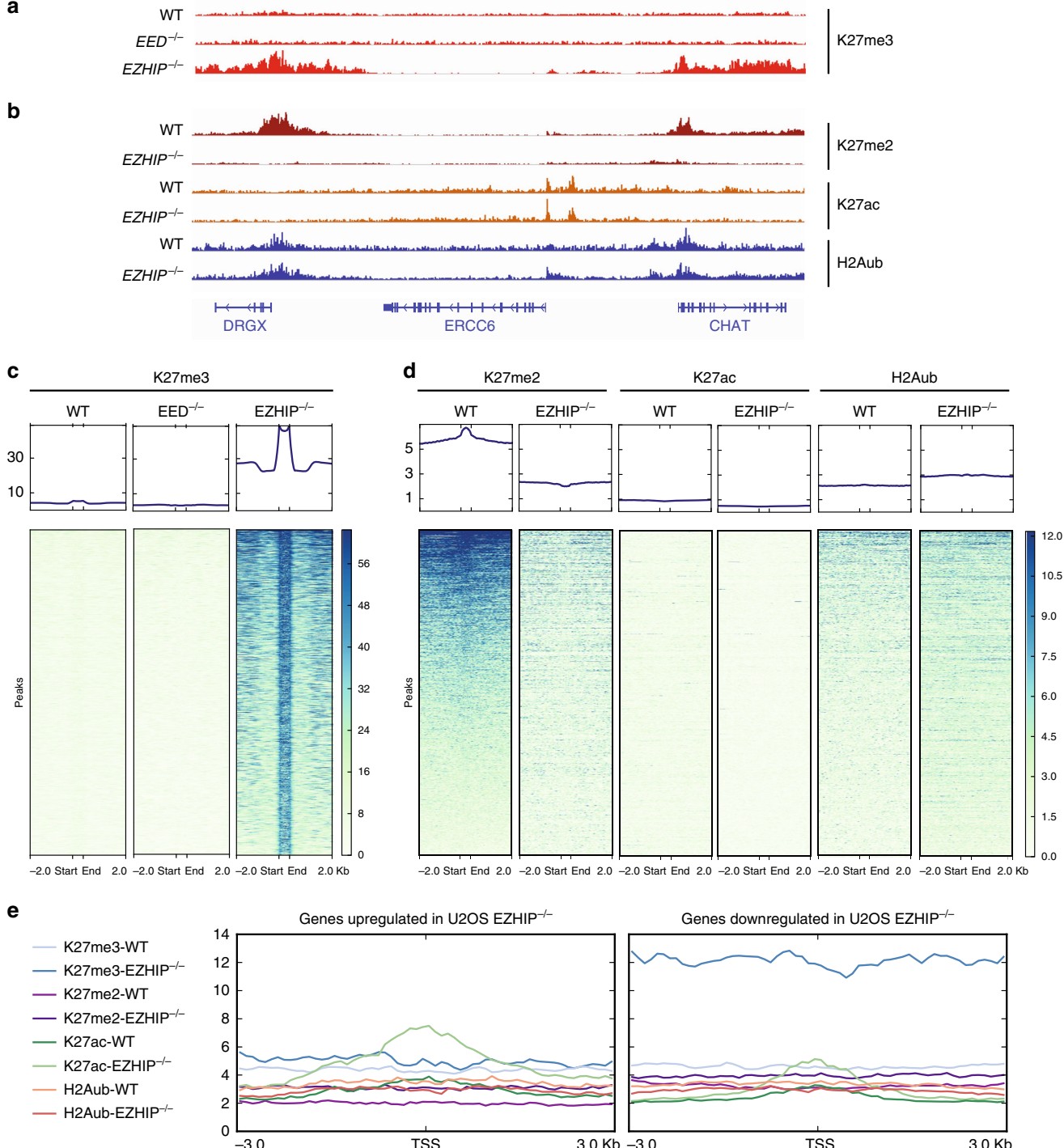

**Fig. 3** EZHIP-mediated interplay between H3K27me2 and H3K27me3. **a** Genome-browser representation of H3K27me3 enrichment in U2OS WT, *EED−/−*, and *EZHIP−/−*. Duplicates are merged, same scale for all tracks. **b** Corresponding Genome-browser representation of H3K27me2, H3K27Ac, and H2AK119ub in U2OS WT and *EZHIP−/−*. Duplicates are merged. Same scale for all tracks of the same histone modification. **c**, **d** Heatmaps and corresponding cumulative plots showing the enrichment for H3K27me3 (**c**) and, H3K27me2, H3K27Ac, and H2AK119ub (**d**) at the peaks (n = 34220) found to gain H3K27me3 upon deletion of *EZHIP*. **e** Density plots showing the enrichment for H3K27me3, H3K27me2, H3K27ac, and H2Aub around the TSS of genes either upregulated or downregulated upon known out of *EZHIP* as defined in Fig. 2d

staining appeared as a diffuse nuclear staining which overlaps partially with the signal detected for EED (Fig. 4b). Of note, EZHIP staining tends to be excluded from the bright dots detected with the anti-H3K27me2 antibody. Since EZHIP modestly impacts PRC2 binding to chromatin but H3K27me3 deposition is impaired, this suggested that EZHIP may instead

interfere with PRC2 enzymatic activity. To test this hypothesis, we first evaluated whether a titration of purified EZHIP (Supplementary Fig. 4C) inhibited the enzymatic activity of the recombinant PRC2 core complex in a histone methyltransferase assay. However, even at molar excess, EZHIP did not impact the enzymatic activity of PRC2 (Supplementary Fig. 4D). We then

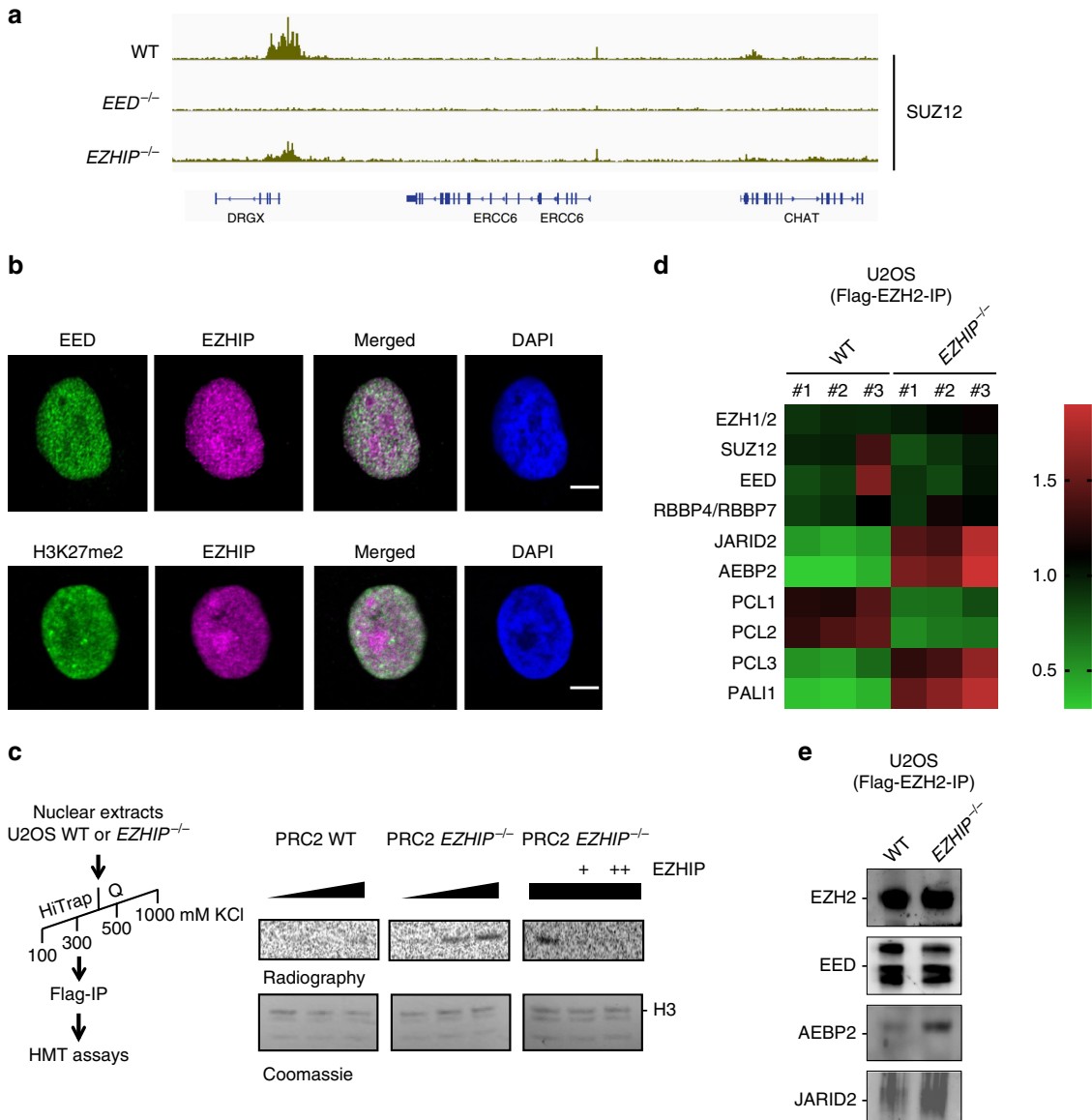

**Fig. 4** EZHIP mitigates PRC2-cofactors interactions. **a** Genome-browser representation of SUZ12 enrichment in U2OS WT, *EED−/−*, and *EZHIP−/−*. Duplicates are merged, same scale for all tracks. **b** Immunofluorescence staining of EED and EZHIP (top) or H3K27me2 and EZHIP (bottom) on U2OS, nuclei are stained with DAPI. Representative results. Scale bar, 2 μm. **c** Left, purification scheme for PRC2. Right, histone methyltransferase (HMT) assay to monitor the enzymatic activity of PRC2 purified from WT (left panel) or *EZHIP−/−* (middle panel) U2OS cells (titration: 1, 2, 5 ×) on native nucleosomes. Right panel, same assay as previously, but this time titrating recombinant hEZHIP on PRC2-purified from *EZHIP−/−* U2OS cells (PRC2 quantity 5 ×). The upper panels are autoradiography, and the lower panels are the corresponding SDS-PAGE coomassie staining. Representative image. **d** Quantification of EZH2-Flag IP through mass spectrometry (iBAQ values). Heatmap representing the Log2-transformed median centered values. Horizontal axis: U2OS WT and *EZHIP−/−*, n = 3. Vertical axes: PRC2 components. Values are normalized on iBAQ values from untagged U2OS WT and *EZHIP−/−*. **e** EZH2-Flag Co-IP from nuclear extracts either WT or *EZHIP−/−*, and probed with antibodies against EZH2, EED, AEBP2, or JARID2

reasoned that EZHIP might regulate PRC2 activity only in the presence of its cofactors. To test this hypothesis, we purified the core PRC2 and its cofactors from U2OS and U2OS *EZHIP−/−* cells that stably overexpress a Flag-tagged version of EZH2 (Supplementary Fig. 4D). EZH2 was immunoprecipitated, and further purified through an ion-exchange column before monitoring its activity on native histones. While we observed very low methyltransferase activity toward H3 with PRC2 purified from WT cells, the complex purified from U2OS *EZHIP−/−* was much more active (Fig. 4c left vs. central panel). Furthermore, in contrast to our observation, with the recombinant core PRC2 complex, the titration of EZHIP on PRC2 purified from U2OS *EZHIP−/−* inhibited PRC2 enzymatic activity (Fig. 4c right panel). These

results suggest that EZHIP might regulate PRC2 by mitigating its interaction with its cofactors. To test this hypothesis, we analyzed PRC2 interactome by mass spectrometry, depending on *EZHIP* expression status. Overall, PRC2 displayed the same composition (Supplementary Fig. 4F); however, the stoichiometry of the cofactors appeared substantially different in the absence of EZHIP (label-free quantification based on iBAQ). Namely, several cofactors—AEBP2, JARID2, and PALI1—were present at a higher stoichiometry in the IPs from *EZHIP−/−* cells. (Fig. 4d). We confirmed this result by co-IP/WB investigating the interaction of AEBP2 and JARID2 with EZH2 in IPs performed with nuclear extracts prepared from U2OS wild-type or *EZHIP−/−* cells (Fig. 4e). Our results suggest that EZHIP does not prevent PRC2

binding to chromatin, but limits the stimulatory action of cofactors, such as AEBP2 and JARID2 on its enzymatic activity.

**_Ezhip_−/− males are fertile despite H3K27me3 increase**. To study the role of EZHIP in a more physiological environment, we generated a knockout mouse model in which a CRISPR-Cas9-induced deletion of 1.5 Kb removes most of the gene body (Supplementary Fig. 5A). Accordingly, _Ezhip_ mRNA and protein were absent from testis and from ovaries (Supplementary Fig. 5B–D), two organs where _Ezhip_ is preferentially expressed. Of note, expression of the genes flanking _Ezhip_ (_Nudt10_ and _Nudt11_) were unaffected by the deletion (Supplementary Fig. 5B, D). _Ezhip_ mice (−/− or −/Y) did not show any overt developmental defect, with adults appearing undistinguishable from the wild-type.

We first investigated the expression of _Ezhip_ during spermatogenesis in the different subpopulations of germ cells sorted from adult mice based on staining for α6-integrin, the tyrosine kinase receptor c-Kit, and DNA content, as previously described[41,42]. _Ezhip_ was mostly expressed in spermatogonia (α6-integrin positive, Supplementary Fig. 5E). Its expression was very low in spermatocytes I and II, consistent with the global transcriptional inactivation of the X chromosome at these stages[43], in contrast to _Ezh2_ expression, which increases at the final stages of differentiation (4n, 2n and n; Supplementary Fig. 5E).

We then tested whether deletion of _Ezhip_ could enhance H3K27me3 deposition during spermatogenesis, as it does in U2OS cells. For this, we probed nuclear extracts from whole testes of adult mice by western blot. As shown in Fig. 5a, amounts of H3K27me2 and me3 increased by about twofold in the absence of EZHIP, whereas other histone marks remain unchanged. Consistent with our previous observations, this effect was not due to a direct effect on the protein accumulation of PRC2 core components (Fig. 5a bottom panel).

To identify the cellular origin of this H3K27me2/me3 upregulation, we performed immunofluorescence on testis sections. Triangle-shaped somatic Sertoli cells—identified by the presence of two satellite chromocenters in their nuclei after DAPI staining—were strongly positive for H3K27me3 in both WT and KO condition (Fig. 5b, yellow star;[44]). In contrast, germ cells—identified by expression of the germ cell marker TRA98—displayed much stronger H3K27me3 signal in _Ezhip_ −/Y mice compared with WT littermates (Fig. 5b, yellow arrows). This suggest that EZHIP does not regulate H3K27me3 deposition in somatic cells of the testis. Indeed, in _Dnmt3l_ mutant testes that are germ cell free[45], no consistent variation in H3K27me3 patterns was observed in presence or absence of EZHIP (Supplementary Fig. 5F).

To evaluate the functional consequences of aberrant H3K27me3 deposition, we profiled gene expression of α6+c-kit- undifferentiated spermatogonia in WT and _Ezhip_ −/Y mice. Gene expression was moderately affected by the absence of EZHIP: about 125 genes differentially expressed (FDR < 0.05, Fig. 5c), the majority of which were downregulated. To determine the impact on spermatogenesis, we analyzed the different germ cell subpopulations from WT and _Ezhip_ −/Y testes by cell cytometry. The relative sizes of these subpopulations were unaffected (Fig. 5d; Supplementary Fig. 5F), in agreement with the normal testis-to-body weight ratio and normal fertility of _Ezhip_ −/Y males (Supplementary Fig. 5G). Finally, we evaluated sperm quality through computer-assisted images analysis of spermatozoa. Spermatozoa motility was not substantially affected, although spermatozoa from _Ezhip_ −/Y males showed slightly less progressive motility, and were a bit more static (Fig. 5e; Supplementary Table 1). While most

histones are replaced by protamine in mature spermatozoa, a small minority carrying various histone modifications including H3K27me3 is retained[46–48]. To determine whether _Ezhip_ deletion impacts this residual H3K27me3, we quantified this mark in the epididymal sperm. Western blot of sperm extracts isolated from _Ezhip_ −/Y mice displayed higher H3K27me3 levels compared with sperm originating from WT animals (Fig. 5h). Whether this upregulation has any functional consequences remains to be investigated, nonetheless, these results confirm the inhibitory activity of EZHIP on H3K27me3 deposition in male germ cells. Interestingly, they reveal that an excess of H3K27me3 is compatible with spermatogenesis and male fertility.

**EZHIP controls H3K27me3 deposition in postnatal oocytes**. In female, classical assembly of chromatin is conserved throughout oogenesis. While the genome-wide deposition of H3K27me3 in PGCs remains to be investigated, H3K27me3 was reported to be progressively restricted during oogenesis to "non-canonical" locations, such as intergenic regions and gene deserts[28]. To assess whether EZHIP could play a role in the regulation of H3K27me3 during different stages of female germline, we first investigated its expression in available data sets in specified germ cells and in postnatal stages of oocyte development. _Ezhip_ is very lowly expressed in migrating PGCs (E9.5), as they entered the genital ridges (E10.5), and as they undergo epigenetic reprogramming (E12.5)[27]. Its expression increases and mirror the one of Ezh2 only in later germ cells ((GCs) E14.5–E15.5) (Fig. 6a, left panel, data from[49])). Interestingly, _Ezhip_ expression is much higher in postnatal oocytes at all stages of growth (Fig. 6a, left and right panels, data from[49,][50]). _Ezhip_ expression drops sharply post fertilization (Fig. 6b). Considering this pattern, we evaluated H3K27me3 levels in postnatal gametes. We performed pre-pubertal female follicles (P17) immunofluorescence against H3K27me3 in wild-type and _Ezhip_−/− conditions. H3K27me3 levels were slightly higher in _Ezhip_−/− primordial follicles compared to WT (Fig. 6c, right panel for quantification). As this difference became more pronounced in secondary follicles (Fig. 6c), we went on investigating H3K27me3 levels at the fully grown oocyte (FGO) stage in adult females. We first harvested germinal vesicle (GV) oocytes from 3-month-old female siblings and stained for H3K27me3 and DAPI; H3K27me3 levels appeared to be around twice more abundant in _Ezhip_−/− oocytes, both in the less condensed chromatin state with no rim surrounding the nucleolus (NSN) and in the fully condensed chromatin state with a DNA-dense rim surrounding the nucleolus (SN) (Fig. 6d; Supplementary Fig. 6A). Of note, this effect was specific to H3K27me3 modifications, as H3K4me3 levels were not lower in NSN oocytes (Supplementary Fig. 6B). Finally, a strong increase in H3K27me3 deposition was also observed in mature MII oocytes from 4-month-old _Ezhip_−/− females (Fig. 5e). We conclude that EZHIP restrains the deposition of H3K27me3 during oocyte maturation.

**Oocyte defects upon deletion of _Ezhip_**. To evaluate the consequences of this global gain in H3K27me3 on gene expression, we first analyzed the transcriptome of a pool of MII oocytes harvested after superovulation of pre-pubertal females (4-weeks old). RNA-seq analysis revealed a very similar transcriptome for the mutant oocytes compared with WT (Fig. 7a). We next investigated whether transcriptomic alterations could appear with aging, as well as if there could be some variability in the transcriptome of individuals oocytes. To this end, we performed single-oocyte-RNA-seq (9 WT and 10 _Ezhip_−/− 4-month-old oocytes). We first ran the comparison between wild-type to mutant in aged oocytes by pooling the single-cell results to mirror

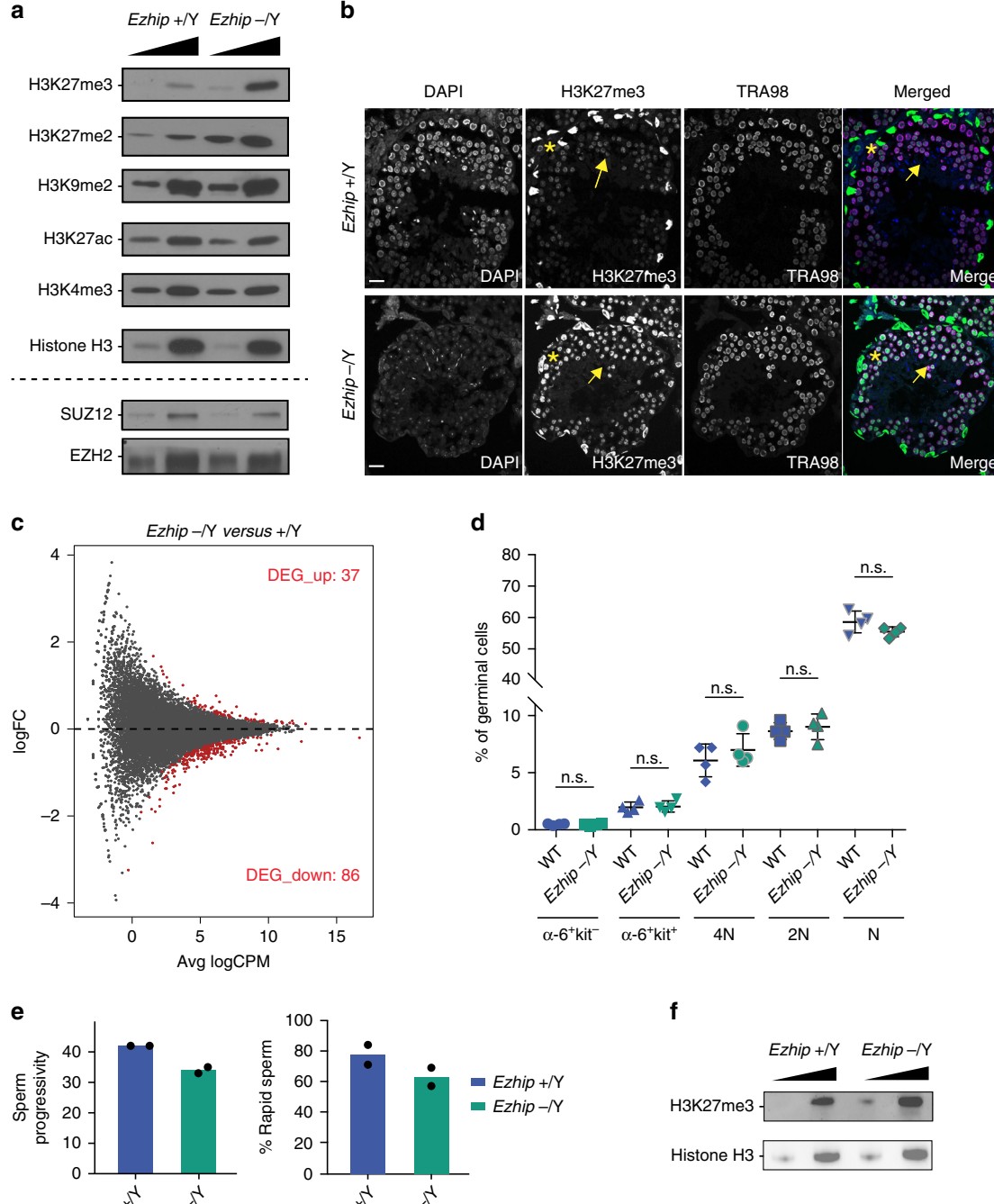

**Fig. 5** Global increase of H3K27me2/3 in male germ cells *Ezhip*−/−. **a** Western blot analysis of H3K27ac, H3K4me3, H3K27me3, H3K27me3, H3K9me2, and H3 on whole testis extracts WT and *Ezhip* −/Y (titration 1, 2.5×). Bottom, same extracts probed for EZH2 and SUZ12. **b** Immunofluorescence detection of H3K27me3 (green) and TRA98 (purple) in testis sections (6-month-old mice). The nucleus is stained with DAPI. Representative results, $n \geq 2$. Scale bar, 25 um. **c** Mean-difference plot between adult male WT and *Ezhip* −/Y sorted spermatogonial population. Differentially expressed genes are highlighted in red (upregulated: 37; downregulated: 86, FDR < 0.05), $n = 2$. **d** Quantification of spermatocyte I (4 N), spermatocyte II (2 N), spermatids (N), and differentiating (a-6 + kit + ), and undifferentiated spermatogonial (a-6 + kit-) by FACS in percent of the total germinal cell population from WT and *Ezhip* −/Y mice (mean ± sem, $n = 4$, Significance: unpaired, non-parametric test of Kologorov–Smirnov). **e** Sperm quality was measured using computer-assisted IVO technologies comparing WT and *Ezhip* −/Y knockout males. Lower left panel: percent sperm progressivity. Lower right panel: percent sperm rapidity (mean, $n = 2$). **f** Western blot analysis of H3K27me3 and H3 on mice sperm extracted from WT and *Ezhip* −/Y animals

our analysis with younger females. We observed that in aged oocytes, the number of significantly differentially expressed genes remains limited, although the comparative expression pattern appears more dispersed (Fig. 7b). This prompted us to determine whether there could be some heterogeneity in terms of gene expression among the oocytes as observed for global H3K27me3

level (Fig. 6d, e). To address this question, we performed principal component analysis of single oocytes. This revealed that while most of the oocytes (regardless of *Ezhip* expression) clustered together, two *Ezhip*−/− oocytes, originating from distinct mice, were clear outliers (Fig. 7c). One of the top genes differentially expressed comparing the outliers and the rest of the

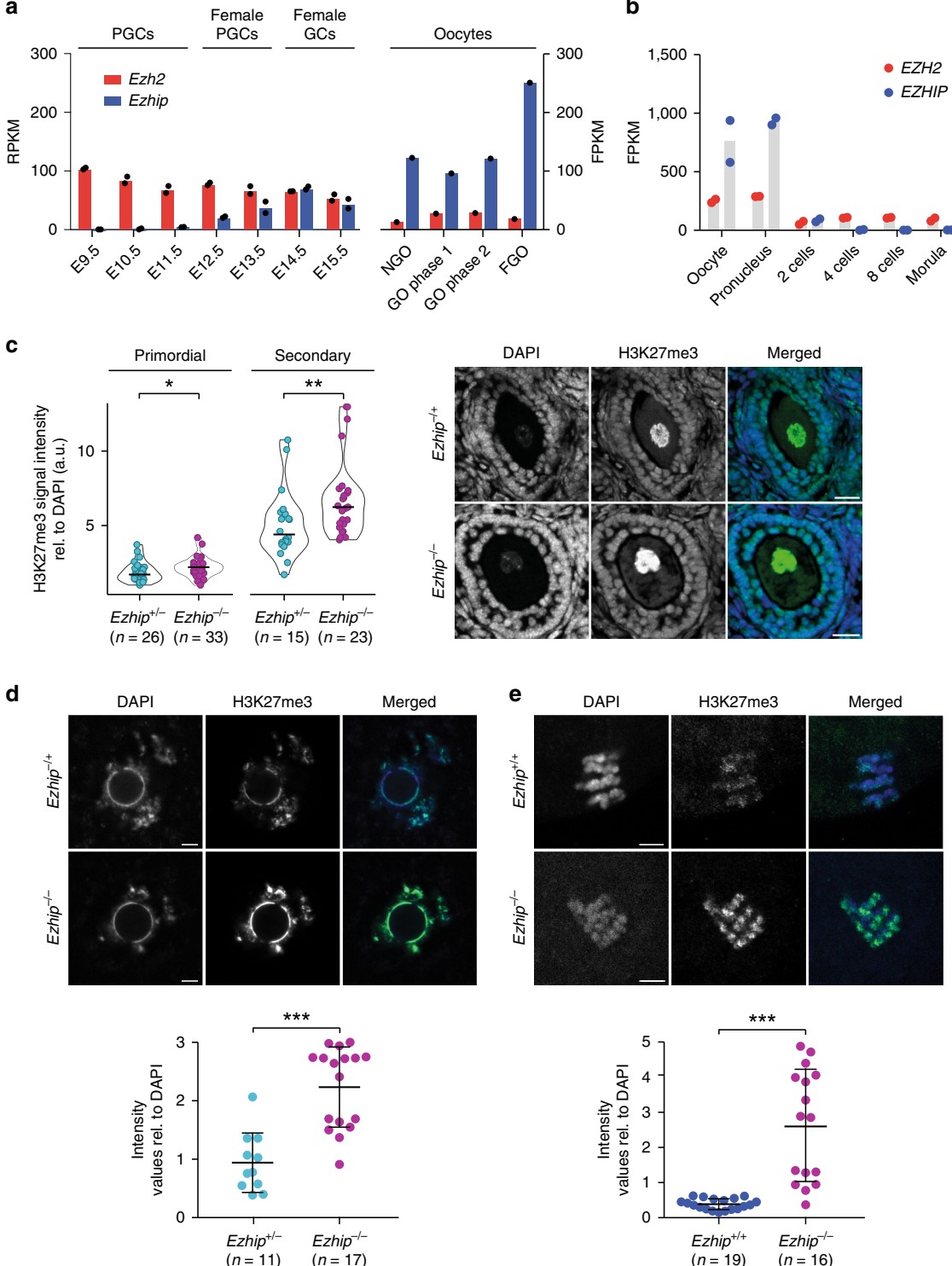

**Fig. 6** Mature oocyte *Ezhip*−/− displays an altered epigenetic landscape. **a** *Ezhip* and *Ezh2* expression (RPKM and FPKM) in germ cells during embryonic development and in oocytes isolated at different stages of follicular growth (PGC primordial germ cells, GC germ cells, NGO non-growing oocyte, GO growing oocyte phase I (8–14dpp) and phase II (15dpp), FGO fully grown oocytes; the data extracted from GSE94136 & GSE70116). **b** Single-cell RNA-seq *Ezh2* and *Ezhip* expression data on early embryo developmental phases (oocyte, pronucleus, two cells, four cells, eight cells, and morula, data from GSE80810). **c** Quantification of H3K27me3 levels in P17 old females primordial and secondary follicles detected by immunofluorescence. Right, H3K27me3 intensities are normalized to DAPI. Left, representative image of secondary follicles stained with DAPI, H3K27me3, and merge, *Ezhip* + /− vs. *Ezhip*−/− . Scale bars, 30 µm. **d** Quantification of H3K27me3 levels by immunofluorescence in mouse surrounded nuclei (SN) GV oocytes. Top representative picture, bottom quantification. Scale bars, 5 µm. **e** Quantification of H3K27me3 levels by immunofluorescence in mature mouse MII oocytes. Top representative picture, bottom quantification. Scale bars, 5 µm. **c**–**e** Mean ± s.d., each dot represents a follicle, *n* indicated on the graph. Significance: unpaired, nonparametric test of Kologorov–Smirnov, \*\*\*$P \leq 0.001$, \*\*$P \leq 0.01$, \*$P \leq 0.05$

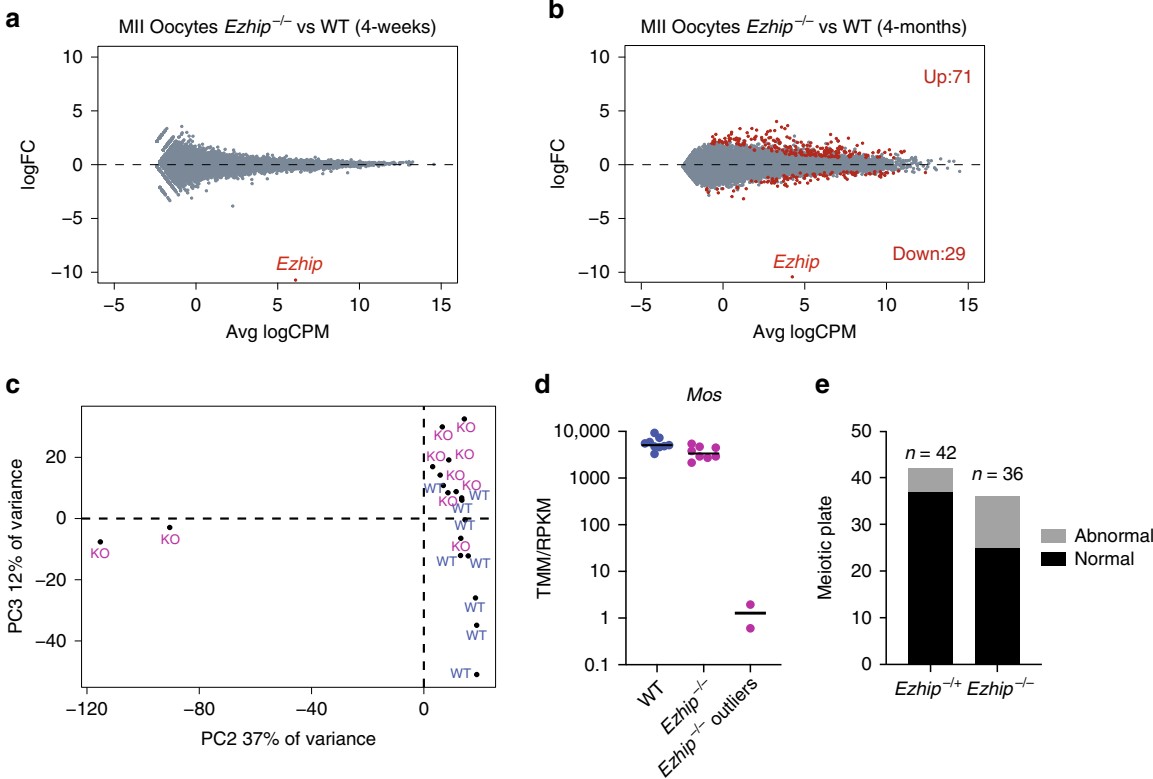

**Fig. 7** Characterization of *Ezhip−/−* oocytes. **a** Mean-difference plot showing log2 fold-change (logFC) expression vs. average log2 counts per million (logCPM) for MII oocytes obtained from 4 weeks *Ezhip + /−* and *Ezhip−/−* superovulated females, *n* = 2 (each *n* is a pool of two individual animals). **b** Expression *vs.* log2 fold-change (logFC) expression vs. average log2 counts per million (logCPM) for MII single oocyte obtained from 4-month-old *Ezhip + /−* and *Ezhip−/−* superovulated females, *n* = 9 for WT and *n* = 10 for Ezhip−/−. Oocytes originated from two different mice for WT and 3 for *Ezhip−/−*. **c** Principal components analysis of each individual oocyte transcriptome. **d** Mos expression according to the single oocyte RNAseq included in (**c**). **e** Chromosome abnormalities evaluated as a proportion of matured MII stages oocytes that exhibited normal alignment of chromosomes on spindle vs. matured MII stages oocytes with lagging chromosomes

Ezhip−/− oocytes was *Mos* (Fig. 7d). Mos has been shown to be required for MAP kinase activation during oocyte maturation, and its deletion impairs microtubules and chromatin organization during the MI to MII transition[51]. Next, we checked the chromosome metaphase plate in MII oocytes, and found that *Ezhip* mutant mice displayed a slight increase in the number of oocytes with lagging chromosomes compared with control (6-week-old; Fig. 7e). Our single oocytes RNAseq and staining of individual oocytes revealed some heterogeneity in oocyte maternal pool as well as in general competence for fertilization. Altogether, our results support a general role for EZHIP in oocyte fitness by regulating H3K27me3 deposition.

**Impaired fertility of *Ezhip* knockout females**. We next investigate whether EZHIP could be involved in the control of follicle maturation. We did not observe any significant differences in the number of primordial, primary, and secondary/antral follicles of pre-pubertal females (P17) regardless of *Ezhip* expression status, indicating that the initial oocyte pool is apparently intact (Fig. 8a, top panels). In contrast, sections from older females (16 weeks) showed a global reduction in the follicle number in the absence of EZHIP (Fig. 8a, bottom panels), although the low number of mature follicles (primary and secondary/antral) at this age was insufficient to reach statistical significance. Collectively, these data suggest a progressive, age related, exhaustion of primordial follicle reserve, from which growing follicles develop. Incidentally, ovaries from *Ezhip−/−* females appeared smaller, with a weight

that was reduced by about 30% compared with wild-type and heterozygous counterparts (Fig. 8b).

In line with these results, we observed that *Ezhip−/−* female mice give rise to fewer progeny. We therefore monitored their fertility by comparing the size and number of litters of WT vs. mutant females (Fig. 8c). Six-week-old WT and *Ezhip−/−* females were mated with a reliably fertile male in the same cage, and monitored daily for 20 weeks (Fig. 8c). All litters were genotyped to assign them to the correct mother; the numbers of mice at birth and at 3 weeks of age (time of genotyping) were similar. However, the total number of pups obtained from *Ezhip−/−* mothers considerably decreased each month, as the females aged (Fig. 8c). This reflected both a reduction in litter number (WT females gave birth to around three litters over a period of 20 weeks while mutant females gave birth to only one litter) and litter size (WT females gave birth to an average of 8 pups/litter while the *Ezhip−/−* average litter size was around 3/4 pups/litter) (Fig. 8d). This result is unlikely to result from developmental defects of the reproductive track since uterine horns appeared normal in adult *Ezhip−/−* females (Supplementary Fig. 7).

We conclude from these experiments that absence of *Ezhip* in oocytes leads to alterations of the epigenetic landscape and is associated to strong reduction in female fertility. Whether the impairment in oocyte pool and its fitness might impair subsequent development of the embryo around and after fertilization remains to be determined.

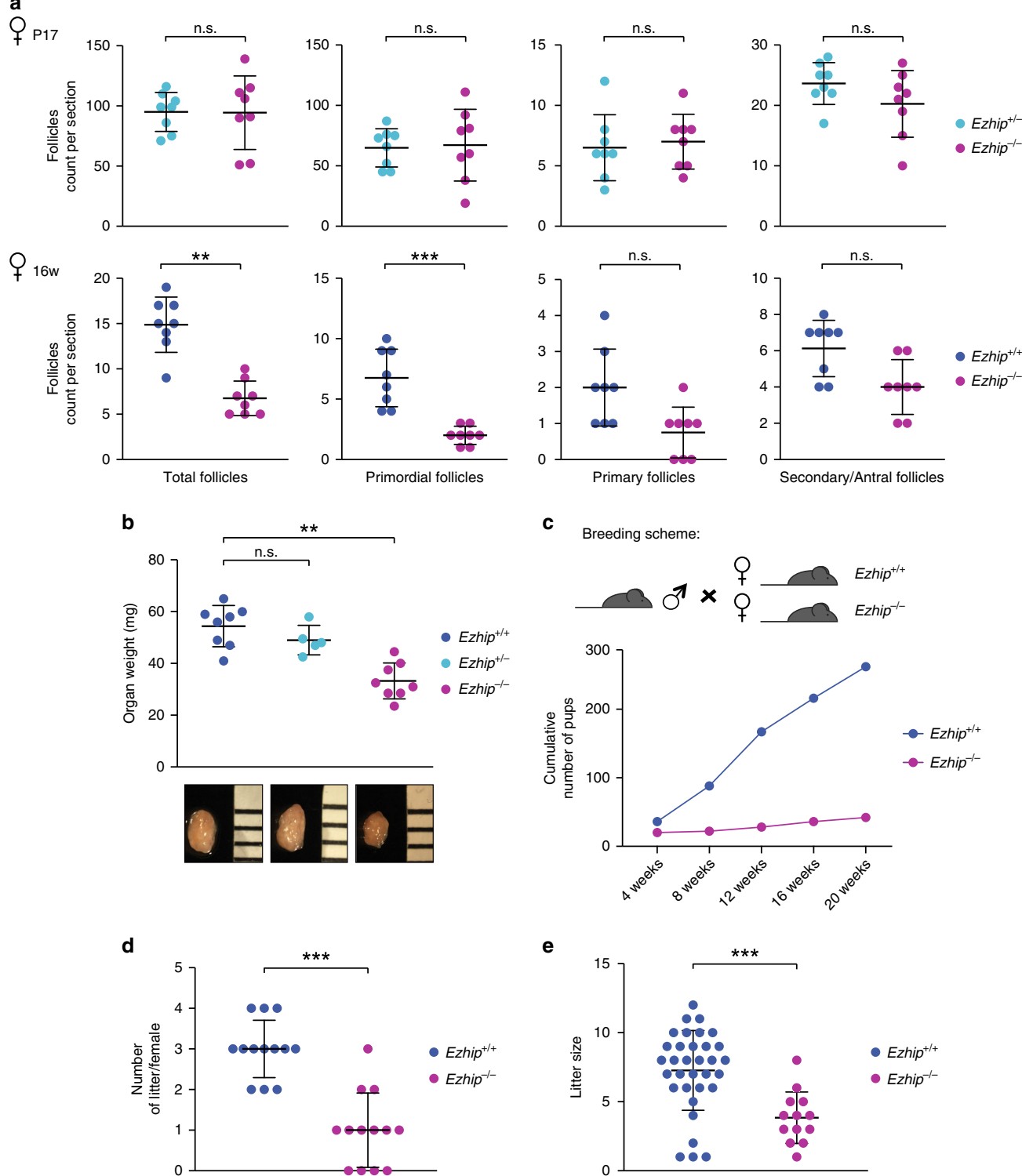

**Fig. 8** *Ezhip* knockout affects adult female fertility. **a** Follicle counting on WT vs. *Ezhip*−/− 2 week-old female slides and 16-week-old, upper and lower panel, respectively (eight slides counted for each genotype, mean ± s.d.). Left panel corresponds to the total number of follicles, then each panel corresponds to a different folliculogenesis step: primordial follicles, primary follicles, secondary/antral follicles. *Y*-axis represents the average follicles number per slide, genotype is indicated in the legend. **b** Average ovaries weight (mg) in adult females (4–5-month-old, mean ± s.d., each dot represents an independent female). **c**–**e** Fertility of WT and *Ezhip*−/− females monitored during 5 months. **c** Cumulative number of pups per genotype depending on time (*n* = 13 cages, breeding scheme represented on top). **d** The number of litters per female during the 20 weeks monitoring. **e** Litter size. **a**, **b**, **d**, **e** Significance: unpaired, non-parametric test of Kologorov–Smirnov, ***P ≤ 0.001, **P ≤ 0.01

## Discussion

Gametogenesis entails significant reprogramming of the epigenome. While histone replacement in spermatogenesis and the progressive loss of DNA methylation during germ cell specification are well documented[27,52], less is known about the regulation of histone posttranslational modifications during this process. Here, we focus on the Polycomb complex PRC2 to investigate this question. We identify an additional PRC2 interacting protein specific to the gonad and showed that it inhibits PRC2 enzymatic activity. Inactivation of this factor leads to a global increase of H3K27me3 during both spermatogenesis and oogenesis. Alteration of the epigenetic content of oocytes leads to a severely compromised fertility.

The PRC2 complex exists in several flavors, depending on which enzymatic subunits it is formed around (EZH1 or EZH2) and depending on which set of cofactors it interacts with[7]. It is known that EZH1 and EZH2 exert redundant functions in spermatogenesis[25], consistent with this redundancy both subunits have a similar interactome in adult mouse testis. Among it, EZHIP contrasts with most of the cofactors identified to date: (i) its expression seems mostly restricted to germ cells, (ii) homologs have only been found in *Eutherians* and it is a fast-evolving protein, (iii) it is a robust inhibitor of PRC2 enzymatic activity, and (iv) it pulls down the entire PRC2 interactome. These last two characteristics are likely linked: it is expected that effective inhibition of PRC2 requires all flavors of PRC2 to be regulated. The poor sequence conservation of EZHIP sequence and its rather disordered structure prediction are more surprising considering that PRC2 and its cofactors are, in contrast, very well conserved. This suggests that the specificity of action of EZHIP on PRC2 could be primarily conferred by the conserved stretch of 13 amino acids. Such a mechanism involving a short-linear motif in direct contact with binding partners (including chromatin modifiers) is a common strategy for parasites such as *toxoplasma* to manipulate the host cellular machineries[53]. It will be particularly interesting to perform structural analyses in order to precisely determine how this interaction occurs, how it interferes with the binding of AEBP2, JARID2, or PALI1 to PRC2, and how this impairs the enzymatic activity of PRC2 without impacting its recruitment to chromatin.

Another interrogation raised by this study is the advantage of expressing an inhibitor of PRC2 to limit H3K27me3 deposition in the gonads rather than downregulating the enzyme itself. We speculate that an inhibitor enables a tighter control over the timing of the reduction in PRC2 activity. Consistent with this possibility, *Ezhip* was recently identified among a set of genes that is expressed in *PGCs*, in response to the developmental DNA demethylation of the germline genome[37]. Of note, its localization on the X chromosome, explains that it remains expressed in oocytes while it is silenced in spermatocytes due to meiotic sex chromosome inactivation. The link between the wave of DNA demethylation and expression of this inhibitor of PRC2 raises the question of whether both processes are functionally related (i.e. whether PRC2 has to be inhibited when DNA methylation is lowered). It will be interesting to map H3K27me3 deposition in *Ezhip*−/− oocytes in order to determine whether it maintains broad enrichment in intergenic regions or at gene deserts[30] and also, whether it could impact on the reestablishment of DNA methylation during oocyte growth.

Recent reports have shown that H3K27me3 on the maternal genome is important for the regulation of allele-specific gene expression[23], and therefore that disrupting PRC2 activity in oocyte through the deletion of *Eed* impairs, post-fertilization, the allelic expression of a subset of genes[54]. Conversely, it is tempting to speculate that PRC2 activity might be limited by EZHIP in order to prevent it from invading genomic regions and thus potentially promoting excessive imprinting. Our results also suggest that excessive H3K27me3 levels resulting from *Ezhip* deletion in testicular germ cells are partially retained in mature spermatozoa. Although it does not seem to impact on the fertilizing properties of the spermatozoa, it will be interesting to determine whether embryos derived from oocytes fertilized with *Ezhip*−/− sperm develop normally. If they do, it would be consistent with the report that paternally inherited H3K27me3 is rapidly erased in the zygote, and carries limited intergenerational potential[28].

Finally, both gain and loss of PRC2 function are a recurrent observation in cancers. While we were completing this study, another publication reported the identification of EZHIP (CXORF67) as an inhibitor of PRC2 in two cancer cell lines (U2OS and Daoy Cells[55]). This demonstrates another means by which cancer cell lines might curtail PRC2 function. Further studies are required to know whether *EZHIP* upregulation might be a recurrent event in cancers and act as a driver of tumor progression. Of note, *EZHIP* has also been involved in gene translocations occurring in endometrial stromal sarcoma, a rare malignant tumor of the uterus[56]. Previous reports revealed frequent fusion between the transcriptional repressor *JAZF1* and the PRC2 core component *SUZ12*, and it was proposed that this fusion could alter PRC2 function[57]. Interestingly, PRC2-cofactors can also be involved in fusions with transcriptional regulators, as it is the case for *PHF1* with *JAZF1*, *MEAF6*, or *EPC1*[58]. Our study extends this observation by showing that the fusion between EZHIP and the nuclear protein malignant brain tumor domain-containing 1 (MBTD1) could result in aberrant PRC2 targeting[58]. It will be important to investigate how these fusions contribute to tumor progression and whether the inhibition of PRC2 could constitute a therapeutic strategy.

## Methods

**Cloning**. m*Ezhip* cDNA clone was obtained from ORIGENE (Ref. MG214772). h*EZHIP* cDNA clone was amplified from HEK-293T genomic DNA. h*EZHIP* mutant 1 (a.a. 1–420), *EZHIP* mutant 2 (a.a.100–503), *EZHIP* mutant 3 (a.a. 200–503), *EZHIP* mutant 4 (a.a. 300–503), *EZHIP* mutant 6 (a.a. 1–450), and *EZHIP* mutant 7 (a.a. 1–395) were generated by PCR and cloned into pMSCV-Hygromycin retroviral vector and/or pCMV4-HA. *EZHIP* mutant 5, depleted of 13AA conserved stretch (a.a.1–503, Δ 394–417) was generated by amplifying the two flanking parts by PCR with overlapping overhangs[59]. *EZHIP* and *Ezhip* cDNA were amplified by PCR and subcloned into pREV retroviral vector (gift form S. Ait-Si-Ali), downstream a 2 × -Flag-2 × -HA sequence and upstream IRES followed by CD25 cDNA.

**Cell lines**. U2OS (ATCC) and HEK-293T (Invitrogen) cell lines were grown according to the manufacturer's instructions. Cell lines were tested for the absence of mycoplasma every month. All transfections were performed using PEI (polyethylenimine) and 150 mM NaCl at 6:1 ratio to DNA. U2OS *EED*−/− cell line was generated by co-transfecting (i) gRNA targeting *EED*, (ii) hCas9, and (iii) a targeting cassette bearing Hygromycin resistance flanked by 1 kb sequences homologous to *EED* locus. Hygromycin B clone selection was performed at 0.2 mg/ml. U2OS *EZHIP*−/− cell line was generated by the same strategy with a targeting construct conferring puromycin resistance (selection was performed at 0,5 mg/ml). Selected U2OS *EZHIP*−/− clone has also undergone NHJ reparation with around 20 bp deletion at the N-terminal part of the sequence. Rescue experiments on U2OS *EZHIP*−/− cell line was performed by infection with retroviral vectors expressing *EZHIP* FL or mutants stably selected with Hygromycin B 0.2 mg/ml.

HeLa-S3 cells (gift form S. Ait-Si-Ali) were grown in the DMEM. pREV retroviruses are produced by transfecting of 293T-Phoenix cell line (gift form S. Ait-Si-Ali) and collecting supernatant after 60 h. HeLa-S3 cells were infected by incubation with viral supernatants for 3 h at 37 °C. Infected cells were then selected by FACS sorting using an anti-CD25-FITC-conjugated antibody and following manufacturer instructions (BD Biosciences 553866). Expression of the recombinant proteins were assessed by WB analysis of nuclear extracts.

**Retroviral production**. Production of pMSCV-Hygromycin retroviral vectors was performed in 293T cells. Transduction was performed by incubating the cells with viral particles mixed with Polybrene (final concentration, 8 µg/ml) for 3 h at 37 °C and subsequently selected with Hygromycin B was added at 0.2 µg/ml.

**Nuclear extract and Flag-IP.** For nuclear extract preparation, cells were incubated with buffer A (10 mM HEPES pH 7.9, 2.5 mM MgCl₂, 0.25 M sucrose, 0.1% NP-40, 0.5 mM DTT, 1 mM PSMF) for 10 min on ice, centrifuged at 7000 xg for 10 min, resuspended in buffer B (25 mM HEPES pH 7.9, 1.5 mM MgCl₂, 700 mM NaCl, 0.5 mM DTT, 0.1 mM EDTA, 20% glycerol), sonicated and centrifuged at 21,000 xg during 15 min. For immunoprecipitation 1 mg of nuclear extract diluted in BC0 to a final salt concentration of 250 mM was incubated with 125 µl of Flag M2 Beads (SIGMA-ALDRICH-A4596), washed three times with BC250 (50 mM Tris pH 7.9, 250 mM KCl, 2 mM EDTA, 10% Glycerol, and protease inhibitors), and eluted with 0.2 M glycine pH 2.6.

**Protein gel and immunoblotting.** Nuclear extracts were fractionated by SDS-PAGE and transferred to the nitrocellulose by semi-dry transfer (Bio-Rad). The membrane was blocked with 5% nonfat milk in PBS-0.1% Tween-20 for 60 min, the membrane was washed once with the same buffer and incubated with antibodies at 4 °C overnight. Membranes were washed three times for 5 min and incubated with either a 1:5000 (mouse)/1:10000 (rabbit) dilution of horseradish peroxidase-conjugated secondary antibodies or with fluorophore-conjugated 1:5000 dilution of Starbright700 (Biorad) for 2 h at room temperature. Blots were washed three times and developed with SuperSignal™ West Pico PLUS Chemiluminescent Substrate (Thermo Fisher). Immunoblots incubated with fluorescent secondaries antibody were visualized using the BIORAD ChemiDoc MP. Raw data for western blot are provided in Supplementary Figs. 8–14.

**Mass spectrometry analysis.** Affinity purifications and liquid chromatography coupled to tandem mass spectrometry (LC–MS) analysis was performed with the same protocol for either testis nuclear extracts, HelaS3 nuclear extracts or U2OS[60]. In brief, nuclear extracts were subjected to a single step Flag-immunoprecipitation (IP) in triplicate (ipFLAG). Control IPs were performed on extracts not expressing the Flag-tagged protein[61]. Nuclear extracts from the Flag-tagged cell line were also incubated with beads lacking Flag antibody. Thus, nine pulldowns were performed in total, three specific pulldowns and six control pulldowns. Precipitated proteins were subjected to on-bead trypsin (Promega) digestion. Peptides were extracted and analyzed by nanoLC-MS/MS using an EASY nLC 1000 system (Thermo Scientific) coupled to an Orbitrap Fusion mass spectrometer or Q Exactive mass spectrometer (Thermo Fisher Scientific). Samples were desalted and concentrated on c18 STAGE tips. After elution the peptides were separated on a C18 column (75 µm i.d. x 30 cm, packed with C18 Reprosil -Pur, 1.9 µm, 100 Å; Dr. Maish) equilibrated in solvent A (water containing 0.1% HCOOH). Bound peptides for the fusion were eluted using a linear gradient of 140 min (from 9 to 32% (v/v) of solvent B (80% MeCN, 0.1% HCOOH), and then is increased to 95% of solvent B, at 250 nl min flow rate and an oven temperature of 40 °C. Samples measured on the QE were eluted using a 120 min (from 9 to 32% (v/v) of solvent B (80% MeCN, 0.1% HCOOH), and is increased to 95% of solvent B, at 250 nl min flow rate and an oven temperature of 40 °C. Fusion samples were acquired in Survey MS scans in the Orbitrap on the 400–1500 m/z range with the resolution set to a value of 120,000 and a $4 \times 10^5$ ion count target. Tandem MS was performed by isolation at 1.6 Th with the quadrupole, HCD fragmentation with normalized collision energy of 35, and rapid scan MS analysis in the ion trap. The MS2 ion count target was set to $10^4$, and the max injection time was 35 ms. Only those precursors with charge state 2–7 were sampled for MS2. The dynamic exclusion duration was set to 60 s with a 5 ppm tolerance around the selected precursor and its isotopes. The instrument was run in top speed mode with 3 s cycles. Samples measured on the QE were collected in a top10 data-dependent acquisition mode with 70.000 resolution for the full scan in the mass range of 300–1650 m/z and a $3 \times 10^6$ ion count target. Fragment ions were acquired by stepped energy of 25. Target was set to $1 \times 10^5$ with an injection time of 120 ms. Only peptides with a charge state of 2–7 were fragmented.

The data processing was done with the MaxQuant software. HelaS3 RAW data were analyzed with MaxQuant version 1.5.0.1 and searched against the uniprot-Human database downloaded from Uniprot (July 2014). Mouse testis RAW data were analyzed with Maxquant version 1.5.0.0 and search against the mouse database downloaded from Uniprot (July 2014). U2OS RAW data were analyzed with MaxQuant version 1.6.1.0 and was searched against the human database downloaded from Uniprot (June 2017). For all the searches, standard settings were used. Trypsin was set as the used enzyme, and methionine oxidation and the N-terminal acetylation were considered as variable modifications, and cysteine carbamidomethyl was set as a fixed modification. Mass tolerance for the precursor was set to 20 ppm. The fragment tolerance for the Fusion was 0.5 Da and for the QE 20 ppm. In all the searches the FDR was set to 0.01.

Identified proteins were filtered for reverse hits and common contaminants. LFQ intensities were log2 transformed, and missing values were semi-random imputed from a normal distribution (width = 0.3 and shift = 1.8) in Perseus. Then proteins were filtered to be detected in all replicates of at least one triplicate experiment. A *t* test in perseus was used to determine the significant outliers.

**Recombinant proteins purification and baculoviruses.** Recombinant EZH2, SUZ12, EED, RBAP48, JARID2 1–530, EZHIP full-length, and mutant proteins were produced in SF-9 insect cells after infection with the corresponding

baculoviruses. Lysates were resuspended in BC300, sonicated and clarified by centrifugation before incubation with either Flag-beads (M2-beads, SIGMA_ALDRICH 4596) and eluted with Flag peptide, or Streptactin–sepharose suspension (IBA, 2-1201-010) and eluted with 2.5 mM Desthiobiotin in BC300. hEZHIP and mutant form baculoviruses were produced accordingly to Bac-to-Bac Baculovirus Expression Systems (Invitrogen) after cDNA cloning into pFASTbac vectors. Recombinant EZHIP proteins were further purified on size exclusion chromatography (S200).

**KMT assay.** KMT assay with recombinant PRC2 and EZHIP proteins were performed with 200 ng of PRC2 alone or in presence of EZHIP, 1 µg of substrates, 4 mM DTT in methylation reaction buffer (50 mM Tris-HCl pH 8.5, 2.5 mM MgCl₂), ³H-SAM, and incubated at 30 °C for 30 min. For KMT assay with PRC2-Flag purified from U2OS WT and *EZHIP−/−*, nuclear extracts were first fractionated on High Trap Q (GE Healthcare) prior to Flag-IP. Nucleosomal substrate for the assay was assembled from 5S 12 repeats DNA[62] and purified HeLa cell histone octamers by salt dialysis through a linear gradient (2.2 M NaCl to 0.4 M NaCl) followed by dialysis against TE solution.

**Antibodies.** Antibodies against EZH1/2, SUZ12, JARID2, and EED were previously characterized[33,63]. RBAP48 mouse mAb (GWB-C12FDE) was purchased from GenWay Biotech; H3 mAb (39163) and H3K27me2 mAb (61435) were purchased from Active Motif. Polyclonal rabbit one against H3 from Cell Signaling Technology (9715); H3K9me2 (ab1220) and H3K27Ac (ab 4729) were purchased from Abcam; H3K27me1 mouse mAb C0321 from Active Motif; H3K27me3 Rabbit mAb C36B11 (9733), H3K4me3 Rabbit mAb C42D8 (9751), Rabbit mAb D7C6X (14129), Rabbit mAb H2AK119ub (8240 S) and mouse mAb10E2 HDAC1 (5356 S) from Ozyme (Cell Signaling Technology). hEZHIP (HPA006128) and mAb Flag-M2 (F1804) were purchased from SIGMA; Anti-Germ cell-specific Rabbit Polyclonal DPP3A/Stella (19878) and TRA98 (Ab82527) Rat monoclonal one from Abcam. For Immunofluorescence, EED was detected with the M26 antibody. Antibody against mEZHIP was raised against the two following synthetic peptides: CAESSRAESDQSSPAG (corresponding to a.a. 91–106) and CAQSAGRNLRPRPRSS (corresponding to a.a. 192–206). Anti-mouse β-TUBULIN was purchased from Invitrogen 32–2600.

Primary antibodies were diluted 1:3000 for WB analysis and 1:250 for Immunostaining.

**Mouse lines.** Mice were hosted in pathogen-free animal facility. All experimentation was approved by the Institut Curie Animal Care and Use Committee (project APAFIS #14570-2018040917413626-v1), and adhered to European and national regulation for the protection of vertebrate animals used for experimental and other scientific purposes (directives 86/609 and 2010/63). EZH2-Flag and EZH1-Flag[33] knock-in mice were generated at the Institute Clinique de la Souris (ICS). The *Ezhip* mouse line was derived by CRISPR/Cas9 engineering of a 1.5-kb deletion spanning the AU022751 locus (Supplementary Fig. 4) in embryos at the one-cell stage, according to published protocols[64]. Of the 13 pups generated, eight carried at least one modified allele. Two founders (N0) carrying the expected 1.5-kb deletion were selected. The absence of in silico-predicted off-target mutations was verified by Sanger sequencing, the two founders were bred with C57B6N mice. Two additional backcrosses were performed to segregate out undesired genetic events, following a systematic breeding scheme of crossing *Ezhip* heterozygous females with C57B6N males to promote transmission of the deletion. Cohorts of female and male mice were then mated to study complete knockout progeny.

**Histological sections and immunostainings.** For histological sections, testis and ovary from either human patients from Curie Institute Pathology Platform or mice were dissected, fixed for 6 h in 4% paraformaldehyde (Sigma), and washed with 70% ethanol according to pathology platform standard protocols. Organs were paraffin-embedded, sectioned (8 µm), and stained with hematoxylin using standard protocols.

For cryosections, testes and ovaries from adult mice (6-month-old males; 2.5 and 5-month-old females) were dissected, fixed overnight in 4% paraformaldehyde at 4 °C, washed in PBS, followed by two consecutive overnight incubations in 15 and 30% sucrose at 4 °C, respectively. Testes were embedded in O.C.T. compound (Tissue-Tek), 8–10 µm-thick sections were cut and spotted onto Superfrost Plus slides (Thermo Fisher Scientific).

For immunofluorescence detection, testis slides were brought to room temperature, blocked and permeabilized for 1 h (10% donkey serum, 3% BSA and 0.2% Triton). For IF on cell lines, slides were fixed with PFA (paraformaldehyde 4%) for 5 min, permeabilized for 5 min (Triton 0.5% in PBS), and blocked for 30 min in 20% goat serum in PBS. Slides were incubated with primary antibodies at 4 °C overnight, followed by three PBS-0.1% Tween-20 washes and 2 h incubation with Alexa Fluor-conjugated secondary antibodies. Slides are washed three times again in PBS-0.1% Tween-20 and incubated with DAPI 1 µg/µl for 5 min. After a quick wash in PBS, slides were mounted with Mounting Media (Life technologies). Images were acquired with a Inverted Laser Scanning Confocal LSM700 UV Zeiss

microscope with a ×40 objective and Z-step in the case of Z-stack scanning. For immunofluorescence on p17 female mice ovary section, antigen retrieval is first performed in 10 mM sodium citrate, 0.05% Tween-20, pH = 6 at 80 °C for 20 min, before proceeding with following steps as indicated above. Images of the section are acquired with a Leica SP8 Confocal microscope, with open pinhole and ×10 objective. For IF on cell lines, images were acquired either with a Leica DM6000B or a Zeiss LSM 800.

**Immunohistochemistry on paraffin-embedded human samples.** Slides were baked 1 h at 65 °C, before deparaffinization and hydration in xylene and graded ethanol to distilled water. Endogenous peroxidase was blocked for 5 min in 1.5% $H_2O_2$ in methanol, antigen retrieval step was performed by boiling slides for 20 min in "Antigen unmasking solution" (Vector Laboratories) and cooling down 1 h at RT. Slides are quickly washed in PBS for 5 min, blocked in PBS-2% BSA-5% FBS 1 h at RT, and incubated overnight at 4 °C in a humid chamber with primary antibody. After washing three times for 5 min in PBS, slides were incubated with biotinylated secondary antibody (Vector Laboratories) for 30 min. Slides were washed in PBS three times for 5 min, before incubating with the ABC substrate for 30 min at RT. After washing again with PBS, DAB was prepared according to the manufacturer instruction (Vector Laboratories), and the staining reaction monitored from 1 to 5 min. Slides were stained with H/E following standard methods, dehydration steps from 90% ethanol solution to xylene is performed, and slides were mounted in VectaMount permanent Mounting Media (Vector Laboratories).

**Analysis of mice testis cell populations.** Testicular single-cell suspensions were prepared from 2–3-months-old from WT and *Ezhip*−/− mice. The albuginea was removed, and the seminiferous tubules were dissociated using enzymatic digestion by collagenase type I at 100 U/ml for 25 min at 32 °C in Hanks' balanced salt solution (HBSS) supplemented with 20 mM HEPES pH 7.2, 1.2 mM MgSO$_4$, 1.3 mM CaCl$_2$, 6.6 mM sodium pyruvate, 0.05% lactate. Next, a filtration step was performed with a 40 -μm nylon mesh to discard the interstitial cells. After HBSS wash, tubules were further incubated in Cell Dissociation Buffer (Invitrogen) for 25 min at 32 °C. The resulting whole-cell suspension was successively filtered through a 40 -μm nylon mesh and through a 20 -μm nylon mesh to remove cell clumps. After an HBSS wash, the cell pellet was resuspended in incubation buffer (same as previously plus glutamine and 1% fetal calf serum). Cell concentrations were estimated using Tryptan Blue staining (>95% viable cells).

Hoechst staining (5 μg/ml) of the cell suspensions was performed accordingly to previous report[41,65]. Cells were labeled with anti-β2m-FITC (Santa Cruz), anti-α-6 integrin-PE (GoH3), and anti-CD117 (c-KIT)-APC (2B8) antibodies (BD Pharmingen). For purification of undifferentiated spermatogonia, MACS (Miltenyi Biotech), α-6 integrin positive fraction of cells was obtained using anti-α-6 integrin-PE (GoH3) and anti-PE microbeads according to the manufacturer's protocol. This fraction was labeled with anti-β2m-FITC (Santa Cruz) and anti-CD117 (c-KIT)-APC (2B8), and then sorted. Propidium iodide (Sigma) was added before cell sorting to exclude dead cells. Analyses and cell sorting were respectively performed on LSR II and ARIA flow cytometers (Becton Dickinson).

**Mouse sperm quality test.** Adult male mice were euthanized by dislocation, and cauda epididymis was collected post-mortem after carefully removing fat pad. Epididymis was opened and sperm released in IVF media (Vitrolife). 1:100 sperm dilution was loaded on Ivos (Hamilton Thorne machine) and sperm parameters were evaluated by Remote Capture software.

**Mouse fertility evaluation.** Six-week-old WT and *Ezhip*−/− females (*N* = 13 each genotype) were crossed and monitored for 20 weeks. One WT female and one *Ezhip*−/− female mouse were housed with an adult breeder male tested previously. Cages were monitored daily and pup numbers and litters were constantly registered. Adult females were euthanized at the end of the study and gonad morphology analyzed.

**Organ phenotypic analysis.** Adult males and females starting from 3/4-months old have been euthanized, ovaries/testis collected, and individually weighted. The whole-organ weight has been considered for comparison among the three genotypes for females and testis weight ratio for males.

**Follicle counting.** Sections were prepared as described above:

(1) For p17 mice, follicles were counted from at least 2 sections each organ/genotype (*N* = 4 each genotype). Primary antibody against DPPA3 (Stella) was used to stain germ cells and DAPI staining to stain nuclei. Different types of follicles were classified by surrounding follicular cells shape.
(2) For 16-week-old mice, follicles number were counted from at least 13 sections each organ/genotype (*N* = 3 each genotype). In all, 5 μM paraffin sections have been stained with hematoxylin and eosin. Follicle classification was based on Pedersen and Peters[66]. Ovaries were serially sectioned, and one every three sections was counted. Follicles were counted from 13 sections and represented as average number per slide per genotype. Measurements were done using a Leica Epifluorescence microscope.

**Nuclei isolation and extraction from tissues.** Mice tissues are rapidly extracted in PBS and dounce homogenized (cut into small pieces with scissors, then 6x up-down with loose and 4x with tight pestle) adding sucrose solution 2.2 (sucrose 2.2 M, HEPES 1 M pH 7.6, KCl 3 M, EDTA 0.5 M, spermine 0.1 M, spermidine 1 M, and protease inhibitors). Homogenized is mixed and added onto sucrose solution 2.05 (sucrose 2.05 M, HEPES 1 M pH 7.6, KCl 3 M, EDTA 0.5 M, spermine 0.1 M, spermidine 1 M, and complete set of protease inhibitors added last minute) in ultracentrifuge Beckemann tubes. Spin 45 min, 24 k at 1 °C in a SW28 rotor. Nuclei pellets were resuspended in an equal volume of nuclear lysis buffer (HEPES 1 M, pH 7.6, KCl 3 M, EDTA 0.5 M, glycerol 87%, spermine 0.1 M, spermidine 1 M, NaF 0.5 M, Na$_2$VO$_4$ 0.5 M, ZnSO$_4$ 50 mM, and complete set of protease inhibitors). Samples were snap-frozen in liquid nitrogen.

**GV and MII oocytes isolation from female mice.** Germinal vesicle (GV) stage oocytes were obtained from 12-week-old females. The ovaries were removed, passed in pre-warmed PBS, and transferred to the M2 medium supplemented with 100 μg/mL of dibutyryl cyclic AMP (dbcAMP; Sigma-Aldrich) at 38 °C. The ovarian follicles were punctured with a 21-gauge needle, and GV oocytes (fully grown oocytes exhibiting a centrally located GV) have been washed five times through M2 droplets in order to ensure that dbcAMP is removed. Zona pellucida has been removed by three passages in tyroide acid solution, followed by three washes in the M2 medium. GV oocytes were further washed in PBS and processed for Immunostaining as described below.

Six-week (chromosome abnormality measurement) or 16-week-old mice were superovulated by intraperitoneal injection using 5 IU pregnant mare's serum gonadotropin (PMSG) and 5 IU of human chorionic gonadotropin (hCG) 48 h later. MII oocytes were collected from ovaries of 12-week-old mice in the M2 medium. Cumulus–oocyte complexes were collected from infundibulum at 14 h after hCG treatment, and recovered in the M2 medium. Cumulus cells were dispersed by hyaluronidase (300 IU/mL) for 5 min in the M2 medium, and oocytes were washed twice with M2 medium and left in PBS for 5 min. No zona pellucida removal treatment has been performed.

**Immunofluorescent staining of GV and MII oocytes.** GV or MII Oocytes were fixed for 20 min in PBS containing 2.5% paraformaldehyde (PFA) at room temperature, and washed with 1% BSA-PBS three times. The cells were permeabilized by incubating in 1% BSA-PBS containing 0.5% Triton X-100 for 30 min at RT. After washing with 1% BSA-PBS three times, oocytes were incubated with primary antibodies in 1% BSA-PBS containing 0.1% Triton X-100 O/N at 4 °C. They were subsequently washed once with 1% BSA-PBS-0.1% Triton X-100 and incubated for 1 h in the dark with Alexa Fluor 488-conjugated IgG secondary antibody (dilution 1:250) in the same buffer. DNA was stained twice for 15 min with 4,6-diamidimo-2-phenylindole (DAPI) prior to a 15 min wash in PBS-0,1% Triton X-100. Cells are passed through increasing percentages of glycerol solution for increasing times (2.5% for 5 min–5% for 5 min-10% for 10 min– 20% for 5 min–50% for 15 min–DTG for 15 min) were then mounted on glass slides with ProLong Gold mounting medium (Life Technologies) for sequential Z-stack imaging. Fluorescence was detected using an Inverted Laser Scanning Confocal LSM700 UV Zeiss microscope with a 63x objective and Z-stack scanning. More than eight oocytes were examined for each condition unless otherwise specified.

**RT-qPCR from mouse tissues.** The total RNA was isolated using the Rneasy Mini Kit (Qiagen). cDNA was synthetized using High Capacity cDNA RT kit (4368814-Applied Biosystems), and quantitative PCR was performed with technical triplicate using SYBR green reagent (Roche) on a ViiA7 equipment (Applied Biosystems). At least three biological independent experiments were performed for each assay. Primers sequences are provided in Supplementary Table 2.

**RNA extraction from mature MII oocytes.** Oocytes were incubated in the M2 containing tyroide acid's solution for 2–3 min to remove their ZP (zona pellucida). ZP-free oocytes were carefully washed several times with M2, and were pooled prior to lysis in XB buffer from Arcturus PicoPure RNA isolation Kit (Applied Biosystems). We then added the spike-in control External RNA Control Consortium (ERCC) molecules (Invitrogen). Normalization was performed using ERCC spike in at 1:1,000,000. The purified total RNA concentration was measured using Agilent High Sensitivity RNA ScreenTape on Agilent 2200 TapeStation. First-strand cDNA (from total RNA) was synthesized according to the SMART-Seq™ v4 Ultra™ Low Input RNA Kit protocol (Clontech Laboratories). The PCR-amplified cDNA was purified using SPRI beads (Beckmann Coulter).

**RNA sequencing from mature MII oocytes.** For sequencing, 75 bp paired-end reads were generated using the Illumina MiSeq. Raw reads were trimmed for adapters with cutadapt (1.12) using the Trim Galore! (0.4.4) wrapper (default settings) and subsequently mapped to the complete mouse rRNA sequence with Bowtie2 (2.2.9). Reads that did not map to rRNA were then mapped with STAR (2.5.3a) to the full reference genome (UCSC build GRCm38/mm10) (including the RNA spike-in control sequence ERCC92, Thermo Fisher cat. no. 4456740) using the following parameters: --outSAMtype BAM SortedByCoordinate --runMode alignReads --outFilterType BySJout --outFilterMultimapNmax 20

--alignSJoverhangMin 8 --alignSJDBoverhangMin 1 --outFilterMismatchNmax 999 --outFilterMismatchNoverLmax 0.04 --alignIntronMin 20 --alignIntronMax 1000000 --alignMatesGapMax 1000000 --outSAMprimaryFlag OneBestScore --outMultimapperOrder Random --outSAMattributes All. Gene counts were generated using STAR --quant_mode (uniquely mapped, properly paired reads that overlap the exon boundaries of each gene).

For differential expression analysis, genes were filtered to include those with CPM > 0.2 in at least two samples, and filtered counts were transformed to log2-CPM and normalized with the TMM method. A linear model was fit to the normalized data, and empirical Bayes statistics were computed. Differentially expressed genes for the KO vs. WT were identified from the linear fit after adjusting for multiple testing and filtered to include those with FDR < 0.05 and absolute log2 fold change > 1.

**RNA-seq in spermatogonial stem cells**. Kit- spermatogonial population was isolated by FACS as specified above from adult mice testis WT and *Ezhip* −/Y (pool of three different mice per experiment for each genotype). Sorted cells were resuspended directly in XB lysis buffer from Arcturus PicoPure RNA isolation Kit (Applied Biosystems). The purified total RNA was stored in nuclease-free water, and RNA concentration was measured using Agilent High Sensitivity RNA ScreenTape. cDNA synthesis and library preparation were performed using SMARTer Stranded Total RNA-Seq Kit-Pico Input Mammalian. 100 bp paired-end reads were generated using the HiSeq 2500 platform. Raw reads were trimmed for adapters with cutadapt (1.12) using the Trim Galore! (0.4.4) wrapper (default settings) and subsequently mapped to the complete mouse rRNA sequence with Bowtie2 (2.2.9). Unmapped reads were then mapped with STAR (2.5.3a) to the full reference genome (GRCm38/mm10). Libraries were confirmed to be stranded according to RSeQC after sampling 200000 reads with MAPQ > 30. Gene counts were generated with STAR --quant_mode (uniquely mapped, properly paired reads that overlap the exon boundaries of each gene) using the Ensembl GTF annotation (vM13).

For differential expression analysis, genes were filtered to include those with CPM > 1 in at least two samples. Raw count data were normalized with the TMM method and transformed to log2-CPM. A linear model was fit to the normalized data, adjusting for batch effects, and empirical Bayes statistics were computed. Differentially expressed genes for each KO vs. WT were identified from the linear fit after adjusting for multiple testing and filtered to include those with FDR < 0.05.

**Single-cell RNA amplification**. After cell lysis, RNAs were reverse transcribed into first cDNA strands with UP1 primer (ATATGGATCCGGCGCGCCGTC-GACT(24)) using Super Script III (Invitrogen) at 25 °C for 5 min, and 50 °C for 30 min[67]. Reverse transcriptase was then inactivated by heat treatment at 70 °C for 15 min, unreacted primers were digested by ExoSAP-IT (USB), and RNA was degraded by RNase H (Invitrogen). PolyA tail was added to the first-strand cDNA at its 3′ end by terminal deoxynucleotidyl transferase (Invitrogen). Second strand cDNAs were synthesized using poly(T) primers with a C6-amine-blocked 5′ end anchor sequence (AUP2: ATATGGATCCGGCGCGCCGTCGACT(24)). The double-stranded cDNAs were then amplified by primers with C6-amine-blocked 5′ ends (AUP1: ATATCTCGAGGGCGCGCCGGATCCT(24) and AUP2 primers) for 20 cycles. PCR products were purified using DNA clean & Concentrator (Zymo research). Then, 0.5–5 kb fragments were excised from agarose gels and purify by Zymoclean gel DNA recovery kit (Zymo research). ERCC spike-in (Invitrogen) were added in the lysis buffer (1:1,000,000) to address technical variation and further normalization. Only cells with high quality, based on morphology and amplification yield of housekeeping genes and ERCC sequences, were submitted to sequencing. Single-cell libraries were carried out according to the manufacturer's protocol (Nextera XT, Illumina), and sequencing was performed on an Illumina HiSeq instrument in pair-end 100-bp reads.

**RNA sequencing in U2OS cells**. The total RNA from U2OS cells was extracted with TRIzol. cDNA were generated according to the manufacturer protocols (Illumina). In total, 50 bp single-end reads were generated using the HiSeq 2500 platform. Reads were first mapped to the complete human rRNA sequence with Bowtie2 (2.2.9). Unmapped reads were then mapped with STAR (2.5.2b) to the complete human reference genome (GRCh37/hg19). Gene counts were generated using STAR --quant_mode. Libraries were confirmed to be strand-specific according to RSeQC (2.6.4) after sampling 200000 reads with MAPQ > 30.

For differential expression analysis, genes were filtered to include those with CPM > 1 in two or more samples. Raw count data were normalized with the TMM method and converted to log2-CPM. A linear model was fit to the normalized data, and empirical Bayes statistics were computed for each comparison. Differentially expressed genes for each comparison were identified from the linear fit after adjusting for multiple testing and filtered to include those with FDR < 0.05.

**ChIP-seq**. For ChIPs, cells were fixed in 1% formaldehyde for 10 mn at room temperature, quenched by adding glycine to a final concentration of 0.125 M, rinsed with PBS and resuspended in buffer LB1 (HEPES-KOH pH 7.5 50 mM, NaCl 140 mM, EDTA 1 mM, glycerol 10%, NP-40 0.5%, Triton X-100 0.25% + protease inhibitors). Cells were rocked at 4 °C for 10 min, pelleted and resuspended

in buffer LB2 (NaCl 200 mM, EDTA 1 mM, EGTA 0.5 mM, Tris pH 8 10 mM + protease inhibitors), pelleted again and resuspended in buffer LB3 (EDTA 1 mM, EGTA 0.5 mM, Tris pH 8 10 mM + Protease inhibitors). Sonication was performed on a Bioruptor (Diagenode), 0.5% N-lauroyl-sarcosine was added, and after rocking at RT for 10 min, supernatant was kept. For the IP, chromatin (10 μg) was incubated with antibodies (around 2 μg) overnight in presence of 1% triton and 0.1% sodium deoxycholate. Beads blocked with BSA were added the day after and incubated at 4 °C for 3 h before processing to the washes in RIPA buffer six times (50 mM HEPES pH 7.6, 10 mM EDTA, 0.7% DOC, 1% NP-40, 0.5 M LiCl + protease inhibitors) and once in TEN (10 mM Tris pH 8.0, 1 mM EDTA, 50 mM NaCl). Elution was done in TES (50 mM Tris pH 8.0, 10 mM EDTA, 1% SDS), before reversing the cross-link overnight and incubating the samples successively with RNAse A and proteinase K prior to phenol/chloroform/isoamyl-alcohol DNA extraction.

**CUT and RUN**. For CUT and RUN, we modified slightly the published protocol[68] as follows. We start from 1.10^6 cells of interest (here U2OS) mixed with 50.10^3 *Drosophila* S2 cells used for normalization (spike-in). Cells are washed twice with wash buffer (20 mM HEPES pH 7.5, 150 mM NaCl, 0.5 mM spermidine & Protease inhibitors), and 10 μL of Concanavalin-A-coated bead slurry in binding buffer (20 mM HEPES pH 7.9, 10 mM KCl, 1 mM CaCl₂, 1 mM MnCl₂) is added and rotated for 10 mn at RT. Tubes are put on a magnetic stand and buffer is replaced by 50 μL antibody solution (2 mM EDTA, 1:100 antibody stock in wash buffer supplemented with 0.1% digitonin). Beads are then rotated for 1 h at RT. After washing the beads in wash buffer-0.1% digitonin, they are incubated for 10 mn with pA-MNase diluted in wash buffer-0.1% digitonin. Beads are washed twice with wash buffer-0.1% digitonin and placed in ice–water bath prior to the addition of 2 mM CaCl2 (final concentration) for 30 mn. Digestion is stopped by addition of 2× Stop buffer (340 mM NaCl, 20 mM EDTA, 4 mM EGTA, 0.02 % digitonin, 1:200 RNase A). Chromatin is recovered by incubating the samples on a heat block at 37 °C for 10 mn, centrifuged, and placed on a magnetic stand. The supernatant is then purified on nucleospin Gel and PCR clean up (Macherey Nagel).

**Sequencing and alignment and peak calling**. In total, 100 bp single-end reads were generated using the HiSeq2500 sequencer for H3K27me3 ChIP-seq (WT, dEED, and dCxorf67), and 50 bp paired-end reads for CUT&RUN samples (Suz12, H3K27ac, H2AK119ub, H3K27me3, and IgG). Reads were simultaneously mapped to the human (GRCh37/hg19) and drosophila (dm6) reference genomes with Bowtie2 (2.2.9) using end-to-end alignment with the preset --very-sensitive. PCR duplicates were removed with Picard Tools MarkDuplicates (1.97) and BAM files were filtered to exclude common artifact regions (http://mitra.stanford.edu/kundaje/akundaje/release/blacklists/hg19-human). Peaks were called with MACS2 on combined replicates using the EED KO as a control for the K27me3 and SUZ12 ChIP and IgG control for CUT&RUN samples with the following parameters: -f BAM --gsize hs --broad --broad-cutoff 0.1 --bdg. Reads were counted in bins of length 50 and RPKM normalized and converted to bigWig format using DeepTools bamCoverage (2.4.1). Spike-in normalization: reads mapping to the drosophila genome were counted into 10 kb bins and scale factors were calculated using DESeq2 estimateSizeFactors.

**Reporting summary**. Further information on research design is available in the Nature Research Reporting Summary linked to this article.

## Data availability
All genomic data have been deposited on Gene Expression Omnibus (GEO) under accession number 'GSE130231'. Proteomics data are available via ProteomeXchange with identifier PXD012354. Raw data are provided for all western blots in supplementary Figs 8 to 14. A source file is provided for all graphical representations. Public data sets used in the paper are the following: Fig. 6a (https://www.ncbi.nlm.nih.gov/geo/query/acc.cgi?acc = GSE94136; https://www.ncbi.nlm.nih.gov/geo/query/acc.cgi?acc = GSE70116); Fig. 6b (https://www.ncbi.nlm.nih.gov/geo/query/acc.cgi?acc = GSE80810); Supplementary Fig. S1G; (https://www.ncbi.nlm.nih.gov/geo/query/acc.cgi?acc = GSE89711); Supplementary Fig. S1H; (https://www.ncbi.nlm.nih.gov/sra/?term = SRP057098). All other relevant data supporting the key findings of this study are available within the article and its Supplementary Information files or from the corresponding author upon reasonable request. A reporting summary for this article is available as a Supplementary Information file.

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

## Acknowledgements

Work in the laboratory of R.M. is supported by the labelisation ARC (Fondation pour la Recherche sur le Cancer), the Labex DEEP and the Institut Curie. R.R. was recipient of fellowship from the FRM (Fondation pour la Recherche Médicale). S.D. is a recipient from a PhD fellowship from "La ligue contre le cancer". The R.M. and D.B. labs are supported by ANR funding (Project MATIN). The M.V's lab is part of the Oncode Institute, which is partly funded by the Dutch Cancer Society (KWF). High-throughput sequencing was performed by the NGS platform of the Institut Curie, supported by grants ANR-10-EQPX-03 and ANR10-INBS-09-08 from the Agence Nationale de le Recherche (investissements d'avenir) and by the Canceropôle Ile-de-France. We thank very much Fatima El Marjou and the Mice Transgenic Facility in Curie Institute for the creation of Ezhip mouse model. Mice sperm quality analysis was realized with the help of Céline Daviaud (IVF platform at Curie Institute) and Franck Bourgade (Pasteur Institute). We thank the Histopathology platform (Cochin Institute) for the Tissue sections and the recombinant protein platform (IC) for ProtA-MN purification and production. We acknowledge the Institute Curie imagery facility for the support in microscopy analysis. The Curie Pathology department donated human gonadal samples. We thank Joan Barau, Rafael Galupa, and Edith Heard for stimulating discussions and members of the Margueron lab for valuable comments on the paper.

## Author contributions

R.R. & R.P.P. performed most of the experiments, R.M. conceived the study. K.A. and D. B. helped with oogenesis characterization. S.D. performed IF in U2OS cells. P.L. prepared the library and handle the sequencing. M.G. and P.F. helped with spermatogenesis characterization. M.B. and M.E.T.P. performed in vivo analysis on P17 females. I.B., P.J., and M.V. performed IP/Mass Spectrometry analysis of Ezh1/Ezh2-Flag testis, EZHIP-Flag HeLa and U2OS cell lines. D.Z. and N.S. performed bioinformatic analysis. A.M. and S.A. provided technical support. R.R. and R.M. prepared the paper. All authors contributed to experimental designed and edited the paper.

## Additional information

**Competing interests:** The authors declare no competing interests.

