## [Peer Review File · Nature Communications]

Reviewers' comments:

Reviewer #1 (Remarks to the Author):

Ragazzini et al. identified GFIP as a new PRC2 co-factor in germ cells, which is required for adult female, but not male, fertility. Mechanistically, the authors found that GFIP interacts with PRC2 and inhibits its enzymatic activity by effecting binding of other PRC2 co-factors to the core PRC2 complex. In vivo deletion of GFIP results in an accumulation of H3K27me2/3 during spermatogenesis and oocyte maturation. While spermatogenesis remains unaffected in mice lacking GFIP, adult females have reduced fertility. The results presented here indicate that GFIP is a novel epigenetic factor that controls H3K27me2/3 levels and fertility in adult females.

This a well conducted manuscript convincingly demonstrating the discovery and characterization of a novel PRC2 factor. Before I can recommend publication of this study, the authors should investigate in greater detail the GFIP-mediated molecular mechanisms.

Major points

1. The observation that GFIP negatively regulates PRC2 activity is a central piece in the study and requires further characterization. In contrast to H3K27me3, H3K27me2 levels are present in WT cells. Also, H3K27me2 is also accumulated in GFIP^{-/-} cells. Is there a global gain or a redistribution and gain of this mark genome-wide? H3K27ac levels are reduced in GFIP^{-/-} cells. H3K27ac ChIP-seq should be performed in WT and GFIP^{-/-} cells to evaluate whether sites with reduced H3K27ac levels gain H3K27me3 in GFIP^{-/-} cells. Perhaps important active enhancers are repressed by H3K27me3 in GFIP^{-/-} cells.

2. Recruitment of canonical PRC1 complex depends on H3K27me3. Is RING1B recruited to the new H3K27me3 sites in GFIP^{-/-} cells? What is the genomic-occupancy of RING1B in WT cells with low levels of H3K27me3?

3. Also, is the PRC2 complex co-recruited with GFIP to chromatin in WT cells? ChIP-seq of a PRC2 core component and GFIP should be performed in both WT and GFIP^{-/-} cells (Figure 2D). These experiments are crucial to show that genome-wide co-binding of PRC2 and GFIP to chromatin occurs in sites devoid of H3K27me3.

4. Finally, in Figure 3C, the authors show an increase of the protein levels of both PRC2.1 and PRC2.2 co-factors. CHIP assays should be performed to show a higher occupancy of PRC2 co-factors in GFIP^{-/-} cells compared to WT cells at chromatin.

Minor points

1. What are the mRNA levels of the PRC2 co-factors in testis and ovary (Fig. 1D)?
2. The different staining pattern between GFIP and EZH2 in Figure 1F is not very clear.
3. Prdm14 expression is not analyzed in Figure S1F.
4. It is not clear why Dnmt3l^{-/-} mice were used for in Figure S4F.

Reviewer #2 (Remarks to the Author):

Ragazzini et al: GPIF constrains Polycomb Repressive Complex 2 activity in germ cells

Review

This most recent work from Margueron's group provides an intriguing new glimpse into the regulation of Polycomb activity during the mouse germline development. The authors describe a characterisation of a new component of the PRC2.1 and PRC2.2 complexes. They further show that the newly discovered factor (GPIF) negatively affects the biochemical activity of the PRC2 complex and identify a part of the GPIF protein that is required for this function through a direct physical interaction with the PRC2 complex. Intriguingly, GPIF seems to be highly expressed during the germ line development – the authors thus proceed to generate a mouse knockout model to address a potential biological role of GPIF. This reveal 1) increased H3K27me3 in mouse ko testes, but this without any detectable effect on fertility and 2) increased H3K27me3 levels in mouse ko oocytes with pronounced effect on fertility , including a rapid fertility decline with progressing age.

Overall, the authors present exciting findings that clearly show lots of potential...having said this, I feel rather frustrated as the mouse ko analysis seems rather preliminary. Much more work could (and should) be done to uncover the role of this new Polycomb component and regulator.

These include (but are not limited to):

- Analysis of H3K27me3 and potentially RNA Seq in the 16 weeks old ko oocytes
- Are there any chromosomal abnormalities observed in the MII ko oocytes?

- What happens following fertilisation?
- Where are the ko male fertility test data?
- Ezh1 plays a pronounced role during gametogenesis – this needs to be discussed
- GPIF is an X-linked gene – the expression and function needs to be put into context of X inactivation dynamics in the germ line

Beyond above there is a number of loose ends that need to be fixed:

- Fig2A: does the presence of GPIF effect the abundance of Eed isoforms??
- What are those 500 genes that are misregulated upon GPIF ko in U2OS cells? (Go terms , etc...)
- Fig3A – how were the genes selected for ChIP analysis? These should be the genes affected by the GPIF ko
- Fig 1E,F – this needs to be much bigger, as it is not obvious what the authors are trying to say
- Fig S4F: the H3K27me3 levels are not the same, opposite to what the authors claim
- Fig4E: n=? stats?
- Fig5: definite phase I and II of GO..

Finally a general comment: it would be important to add the number of replicates used in various experiments and stats. Many of the WB/radiographs would benefit from better annotation.

Reviewer #3 (Remarks to the Author):

Thanks to the authors to upload their original data to proteomExchange. However, while this allows the interested and expert reader now to explore the data (which I strongly appreciate), the current reporting on material and methods is a little sloppy. Accurate documentation is the basis in science for replication of experiments and in this case reflects negative on the authors and this study.

The MS analysis cites another publication from some of the authors. This contains all necessary information to repeat the experiment, however it does not contain the specific details asked for by the Reporting summary of Nature Communications. Especially, the data analysis section misses information on e.g. software version of MaxQuant, settings chosen, database used, search engine configuration. I can extract this information now from the provided metadata on proteomExchange. There are more improvements necessary to the Method section, I just comment one a few very

obvious statements here: The authors report to measure on a Velos instrument in the Material and Methods section, but tag a Fusion instrument in the metadata for proteomExchange upload. In fact, the authors measured on a Velos (2 sets) AND on a Fusion (1 set); using different settings on both instruments. To further note, different MaxQuant versions and different database versions were used throughout the study. This is not a problem, but should be mentioned in the material and methods section as required by Nature Communication. There is nothing wrong with the data or the chosen analysis strategy (as far as I can reconstruct from the provided information), but the details and accurateness for documentation is not acceptable yet.

In principle, I have nothing against publication of the manuscript if the biological reviewers suggest this, but the reporting section for the MS data needs to be improved.

First of all, we would like to thank the reviewers for their overall positive comments and constructive criticisms. We performed additional experiments in order to address the specific requests. We believe that this makes the study more robust.

We hope that the reviewers will share our enthusiasm and consider that this manuscript is now suitable for publication.

Below, we provided point-by-point answers to the reviews.

Please note that since another team deposited the name EZHIP for CXorf67/AU022751, we decided to follow this name to avoid confusion in the field. EZHIP therefore replaces GPIF in the entire manuscript.

Reviewers' comments:

Reviewer #1 (Remarks to the Author):

Ragazzini et al. identified GPIF as a new PRC2 co-factor in germ cells, which is required for adult female, but not male, fertility. Mechanistically, the authors found that GPIF interacts with PRC2 and inhibits its enzymatic activity by effecting binding of other PRC2 co-factors to the core PRC2 complex. In vivo deletion of GFIP results in an accumulation of H3K27me2/3 during spermatogenesis and oocyte maturation. While spermatogenesis remains unaffected in mice lacking GFIP, adult females have reduced fertility. The results presented here indicate that GFIP is a novel epigenetic factor that controls H3K27me2/3 levels and fertility in adult females.

This a well conducted manuscript convincingly demonstrating the discovery and characterization of a novel PRC2 factor. Before I can recommend publication of this study, the authors should investigate in greater detail the GFIP-mediated molecular mechanisms.

We really appreciate the positive evaluation given by the reviewer concerning this study.

Major points

1. The observation that GFIP negatively regulates PRC2 activity is a central piece in the study and requires further characterization. In contrast to H3K27me3, H3K27me2 levels are present in WT cells. Also, H3K27me2 is also accumulated in GFIP^{-/-} cells. Is there a global gain or a redistribution and gain of this mark genome-wide?

In order to address the reviewer comment, we performed CUT&RUN to probe H3K27me2 in U2OS WT and *EZH1P*^{-/-}. Results are now included in figure 2. They revealed that the regions gaining H3K27me3 upon deletion of EZHIP are enriched for H3K27me2 in the WT context. This result and the CUT&RUN for SUZ12 support our hypothesis that EZHIP does not interfere with PRC2 binding to chromatin but rather inhibits its enzymatic activity.

H3K27ac levels are reduced in GFIP^{-/-} cells. H3K27ac ChIP-seq should be performed in WT and GFIP^{-/-} cells to evaluate whether sites with reduced H3K27ac levels gain H3K27me3 in GFIP^{-/-} cells. Perhaps important active enhancers are repressed by H3K27me3 in GFIP^{-/-} cells.

To test the reviewer hypothesis, we analyzed H3K27ac in the absence of EZHIP and indeed observed a decrease of H3K27ac while H3K27me2 and H3K27me3 in contrast increase.

We further address this point by performing CUT&RUN to analyze H3K27ac distribution. The result of this experiment is now included in the manuscript.

2. Recruitment of canonical PRC1 complex depends on H3K27me3. Is RING1B recruited to the new H3K27me3 sites in GFIP^{-/-} cells? What is the genomic-occupancy of RING1B in WT cells with low levels of H3K27me3?

We used H2Aub enrichment as a way to evaluate PRC1 recruitment at chromatin. We did not observe dramatic changes upon deletion of EZHIP suggesting either that H3K27me2 can also stabilize PRC1-mediated-H2Aub deposition or that H3K27me3 contribution to H2Aub deposition is limited.

3. Also, is the PRC2 complex co-recruited with GFIP to chromatin in WT cells? ChIP-seq of a PRC2 core component and GFIP should be performed in both WT and GFIP^{-/-} cells (Figure 2D). These experiments are crucial to show that genome-wide co-binding of PRC2 and GFIP to chromatin occurs in sites devoid of H3K27me3.

We have done ChIP-seq as well as CUT&RUN to detect EZHIP at chromatin however we did not get any clear enrichment over the control suggesting either that our experiments did not work or that EZHIP has a rather broad and non-specific binding to chromatin (therefore not significantly different from input). In support of the second hypothesis, EZHIP does not have any defined protein domain expected to mediate specific chromatin interaction. Also, as an alternative method to gauge EZHIP's cell localization, we performed immunofluorescence microscopy. This indicates that EZHIP has a broad distribution in the nucleus which overlaps partially with EED but not that well with the bright dots of H3K27me2 (see figure 3A). We therefore lean toward the hypothesis that EZHIP has a nuclear distribution not necessary dependent on PRC2.

4. Finally, in Figure 3C, the authors show an increase of the protein levels of both PRC2.1 and PRC2.2 co-factors. ChIP assays should be performed to show a higher occupancy of PRC2 co-factors in GFIP^{-/-} cells compared to WT cells at chromatin.

We have done CUT&RUN for JARID2. We observed a low level of enrichment over the IgG in the WT and EED^{-/-} contexts and a robust increase in the U2OS EZHIP^{-/-}. While this could fit with our hypothesis, we remain puzzled by the fact that this increase enrichment of JARID2 is not restricted to Polycomb target regions but also occurs at regions devoid of H3K27me3 (e.g. GAPDH, figure below). This result is not expected for JARID2 considering previous reports showing its colocalization with PRC2. At present, we cannot explain this result and therefore we prefer not to include/publish it until further investigation.

Reviewer Figure 1: Representation of CUT&RUN result (Y axis Ct differential between Jarid2 antibody and IgG), mean +/-S.D., n=2. *DRGX* and *SDHAP* are PRC2 target genes in U2OS while *GAPDH* is highly expressed and devoid of H3K27methylation.

Minor points

1. What are the mRNA levels of the PRC2 co-factors in testis and ovary (Fig. 1D)?

We added a figure (Fig. S1F) showing the expression of PRC2 core components and cofactors in oocytes and spermatogonia.

2. The different staining pattern between GFIP and EZH2 in Figure 1F is not very clear.

We have modified the figure 1E and 1F to make the staining differences more obvious.

3. *Prdm14* expression is not analyzed in Figure S1F.

We added *Prdm14* expression levels as requested.

4. It is not clear why *Dnmt3l*^{-/-} mice were used for in Figure S4F.

We apologize for not making this point clear.

We used these mice as a control to show that *EZH1P* regulates H3K27me3 only in the subpopulation of germ cells that composed the testis. *Dnmt3L*^{-/-} male mice rapidly lose their germ cells, testis are therefore mostly composed of somatic cells. In this context, *EZH1P*'s deletion does not affect H3K27me3 global level.

Reviewer #2 (Remarks to the Author):

Ragazzini et al: *GPIF* constrains Polycomb Repressive Complex 2 activity in germ cells

Review

This most recent work from Margueron's group provides an intriguing new glimpse into the regulation of Polycomb activity during the mouse germline development. The

authors describe a characterisation of a new component of the PRC2.1 and PRC2.2 complexes. They further show that the newly discovered factor (GPIF) negatively affects the biochemical activity of the PRC2 complex and identify a part of the GPIF protein that is required for this function through a direct physical interaction with the PRC2 complex. Intriguingly, GPIF seems to be highly expressed during the germ line development – the authors thus proceed to generate a mouse knockout model to address a potential biological role of GPIF. This reveal 1) increased H3K27me3 in mouse ko testes, but this without any detectable effect on fertility and 2) increased H3K27me3 levels in mouse ko oocytes with pronounced effect on fertility, including a rapid fertility decline with progressing age.

Overall, the authors present exciting findings that clearly show lots of potential...having said this, I feel rather frustrated as the mouse ko analysis seems rather preliminary. Much more work could (and should) be done to uncover the role of this new Polycomb component and regulator.

We are glad that the reviewer considers our findings “exciting” and we agree with him/her that more remain to be done regarding the role of EZHIP in gonads. However, we hope that the reviewer will concede that we cannot address all aspects of EZHIP function within one study. Here, we present the identification of this new PRC2 cofactor, its characterization in a model cell line and report a robust phenotype resulting from its deletion in mice. The full description of the underlying mechanisms will be the subject of subsequent work. Nonetheless, we performed additional experiments that are described below.

These include (but are not limited to):

- Analysis of H3K27me3 and potentially RNA Seq in the 16 weeks old ko oocytes

We thank the reviewer for this suggestion. In order to address his/her request, we have done single oocyte RNA-seq of 4-months-old mice which is now included in the manuscript. In contrast, to our previous observation on pools of young oocytes, this experiment reveals that some oocytes display altered gene expression profile. Hence, the expression of *Mos* -involved in the process of oocyte maturation- is dramatically reduced in those oocytes.

We agree that it will be interesting to precisely map H3K27me3 gains in those oocytes however we were limited by the number of mice to perform this experiment.

- Are there any chromosomal abnormalities observed in the MII ko oocytes?

As suggested by the reviewer, we evaluate chromosomal abnormalities in MII oocytes (after superovulation of 6-weeks-old mice). This experiment shows a slight increase in the number of chromosomal abnormalities and thus support the hypothesis that some *Ezhip* *-/-* oocytes might not be fully functional.

- What happens following fertilization?

This is certainly a very interesting question. Indeed, it is likely that the aberrant deposition of H3K27me3 resulting from the deletion of *Ezhip* could impact gene activation on the maternal genome following fertilization. Yet, as explained above, we

believe that the full characterization of the reduced fertility phenotype is beyond the scope of this manuscript.

- Where are the ko male fertility test data?

We apologize for omitting the raw data, it is now included in table S1.

- Ezh1 plays a pronounced role during gametogenesis – this needs to be discussed

We have modified the discussion accordingly to emphasize this point.

- GPIF is an X-linked gene – the expression and function needs to be put into context of X inactivation dynamics in the germ line

We added another sentence in the discussion to emphasize how *Ezhip* being an X-linked could relate to its pattern of expression during spermatogenesis.

Regarding the females, the inactive X is reactivated in early PGCs before they enter the gonads. Although we don't know the exact timing for the reactivation of *Ezhip*, it is most likely that this gene is reactivated in female embryos when they start their differentiation phase in genital ridges. We therefore expect that in early oocytes the two alleles of *Ezhip* are fully expressed.

Beyond above there is a number of loose ends that need to be fixed:

- Fig2A: does the presence of GPIF effect the abundance of Eed isoforms?

The Figure 2A indeed suggested that EED3/4 could be less abundant in the *EZH1P* KO. We have reproduced this western blot several times and could not confirm this result. We updated the figure accordingly.

- What are those 500 genes that are misregulated upon GPIF ko in U2OS cells? (Go terms , etc...)

We have analyzed the genes misregulated upon *EZH1P* KO and included the result in figure S3E. However, the GO terms did not reveal any striking features.

- Fig3A – how were the genes selected for ChIP analysis? These should be the genes affected by the GPIF ko

We have replaced the figure by a whole genome analysis (CUT&RUN). This new piece of data provides a better evaluation of PRC2 (through SUZ12) enrichment at chromatin and confirms that PRC2 recruitment is modestly impacted by the presence of *EZH1P* (Fig 4A&S4A).

- Fig 1E,F – this needs to be much bigger, as it is not obvious what the authors are trying to say

We have modified the figure according to the reviewer suggestion to better illustrate the results.

- Fig S4F: the H3K27me3 levels are not the same, opposite to what the authors claim

We agree that the presented figure could suggest that H3K27me3 is a bit lower in the absence of *Ezh1p*, however we have done the western blot several times, and, despite slight variations, no consistent trend emerged. Also, the most important conclusion is that in the *dnmt3l*^{-/-} testis, H3K27me3 is not upregulated anymore by the KO of *Ezh1p*.

- Fig4E: n=? stats?

We have analyzed two mice *per* genetic background, therefore n=2.

- Fig5: definite phase I and II of GO.

We have included this information in the figure legend.

Finally, a general comment: it would be important to add the number of replicates used in various experiments and stats. Many of the WB/radiographs would benefit from better annotation.

We have carefully checked the legends to complete the missing information.

Reviewer #3 (Remarks to the Author):

Thanks to the authors to upload their original data to proteomExchange. However, while this allows the interested and expert reader now to explore the data (which I strongly appreciate), the current reporting on material and methods is a little sloppy. Accurate documentation is the basis in science for replication of experiments and in this case reflects negative on the authors and this study.

We apologize for the lack of details of our material and methods section. We have now modified this section to address the reviewer comments; all the requested information should now be included.

The MS analysis cites another publication from some of the authors. This contains all necessary information to repeat the experiment, however it does not contain the specific details asked for by the Reporting summary of Nature Communications. Especially, the data analysis section misses information on e.g. software version of MaxQuant, settings chosen, database used, search engine configuration. I can extract this information now from the provided metadata on proteomExchange. There are more improvements necessary to the Method section, I just comment one a few very obvious statements here: The authors report to measure on a Velos instrument in the Material and Methods section, but tag a Fusion instrument in the metadata for proteomExchange upload. In fact, the authors measured on a Velos (2 sets) AND on a Fusion (1 set); using different settings on both instruments. To further note, different MaxQuant versions and different database versions were used throughout the study. This

is not a problem, but should be mentioned in the material and methods section as

required by Nature Communication. There is nothing wrong with the data or the chosen analysis strategy (as far as I can reconstruct from the provided information), but the details and accurateness for documentation is not acceptable yet.

In principle, I have nothing against publication of the manuscript if the biological reviewers suggest this, but the reporting section for the MS data needs to be improved.

Reviewers' comments:

Reviewer #1 (Remarks to the Author):

I must admit that I was expecting a thorough revised version from the Margueron lab, instead I see the authors did a sloppy revision and wrote a poor point-by-point response letter. The paper needs further revision before I can recommend its publication in Nature Communications.

Although most of my points were addressed, the authors did a very superficial analysis of the new genome-wide experiments. It is not enough to generate couple a correlation matrix (wrongly referred as figure 2 in the point-by-point letter) of a bunch Cut&Run results. Data must be analyzed more carefully and more deeply and crossed with gene expression profiles.

The H3K27ac and H3K27me2 genome-wide results are confusing and not well explained. There is a discrepancy between the H3K27ac and H3K27me2 WBs compared to the Cut&Run results. The authors show by WB increased H3K27me2 signal in EZHIP^{-/-} cells, yet Cut&Run results show a global reduction. Similar discrepancy is observed in H3K27ac analyses.

If antibodies against EZHIP did not work, I would have strongly appreciated an EZHIP ChIP-seq attempt using a tagged version of EZHIP in EZHIP^{-/-} cells.

Reviewer #4 (Remarks to the Author):

Ragazzini et al. present exciting new work characterising a new PRC2 complex member EZHIP. They present quite detailed characterisation in a model cell line and create a knockout mouse model, which has an intriguing phenotype.

In my view (as a new reviewer added at the rebuttal phase), the authors have dealt with the majority of the previous reviewer's comments, and added extra data that has improved the manuscript.

While I agree that a full description of the underlying mechanisms is too high a bar (especially given the novelty of the findings) I do feel that some further basic characterisation should be included:

1. PGC expression: Given that the authors have established a role for EZHIP in regulating H3K27me3 levels in mouse germ cells, I think it would be helpful to include a fuller characterisation of gene expression in PGCs. Is it expressed in newly specified PGCs, during migration or only after genital ridge colonisation? Ideally this could be shown by immunostaining at the relevant stages (with knockout controls), however it may be possible to mine RNA-seq datasets that already exist in the literature (please note I do not think that Supp Figure 1G is sufficient in this regard). Although EZHIP is one of the so-called germline-reprogramming-responsive genes, it is possible the gene is lowly expressed prior to its reprogramming associated activation.

2. H3K27me3 in PGCs: At what developmental stage is the first detectable increase in H3K27me3 observed in EZHIP knockout germ cells? Are there already elevated levels in PGCs? Please note, I'm sure the authors are keen to characterise epigenetic reprogramming in PGCs and in the zygote in these knockouts – and I think it is reasonable to save this for a follow up study. However, I do not think establishing when the epigenetic mark regulated by EZHIP is first affected during PGC/germ cell development, is an unreasonable request. This is especially relevant given the statement in the abstract 'a novel functional player in the comprehensive chromatin remodelling that occurs in the gonad'.

3. H3K27me3 in 4month KO oocytes: I did not think the request by Reviewer #2 for analysis of H3K27me3 in 4month KO oocytes was wholly unreasonable – and the authors should consider whether this could be done. Perhaps, if my points 1 and 2 are adequately addressed, this could be avoided.

With these additions I would consider the paper ready for publication in Nature Communications:

Minor points:

1. Abstract line 45: Evidences?
2. Line 114: Mammalian seems a broad statement here. Is this based on the expression shown in human also? It is hard to argue function based on expression only and if a broader statement is to be made about mammals I would include germ cell expression data in a broader range of species.
3. Line 286-7: Does the deletion remove the conserved amino acid region?
4. Line 325: 'Spermatozoa motility felt within normal standards': This does sound like a somewhat subjective measure? Unless perhaps there is a typographical error?

Reviewer #1 (Remarks to the Author):

I must admit that I was expecting a thorough revised version from the Margueron lab, instead I see the authors did a sloppy revision and wrote a poor point-by-point response letter. The paper needs further revision before I can recommend its publication in Nature Communications.

Although most of my points were addressed, the authors did a very superficial analysis of the new genome-wide experiments.

It is not enough to generate couple a correlation matrix (wrongly referred as figure 2 in the point-by-point letter) of a bunch Cut&Run results.

We are pleased to know that most of the points raised by reviewer #1 are addressed.

We apologize for the typo referring to figure 2 whereas it was figure 3.

We are surprised by the reviewer comment since our analysis included obviously more than a correlation matrix. Indeed, Fig. 3A&B show screenshots of a representative region targeted by PRC2, Fig. 3C&D include heatmaps illustrating changes in chromatin landscape at the regions that gain H3K27me3 upon knockout of EZHIP as well as the corresponding density plots. Finally, supp Fig. 3 shows the overlap between the H3K27me3 peaks in the WT *versus* EZHIP KO condition.

Nonetheless, we have now added Fig 3E and Supp Fig. 3D, and edited the Fig. 3C&D, to address his/her criticism.

Data must be analyzed more carefully and more deeply and crossed with gene expression profiles.

We removed the correlation between gene expression profiles and H3K27me3 enrichment that was included at the first submission as we thought the result to be somehow expected.

Fig. 3E is now showing the chromatin changes occurring at the set of genes either upregulated or downregulated in the absence of EZHIP.

We also modified the manuscript accordingly.

Line 230 to 240.

The H3K27ac and H3K27me2 genome-wide results are confusing and not well explained. There is a discrepancy between the H3K27ac and H3K27me2 WBs compared to the Cut&Run results. The authors show by WB increased H3K27me2 signal in EZHIP^{-/-} cells, yet Cut&Run results show a global reduction. Similar discrepancy is observed in H3K27ac analyses.

We realize that the reviewer is confused regarding the content of the Fig. 3C&D although it was explained both in the result section and in the legend.

In order to avoid such problem with the readers, we have modified the text and added the Supp Fig 3D. This supplementary figure shows that, as expected from the global level of H3K27me2 shown by western blot, we have an increase in the number of peaks detected for H3K27me2 in the EZHIP^{-/-} context. Nonetheless, at the peaks that gained H3K27me3 which are shown in Fig. 3C&D, we have a decrease of this mark probably corresponding to its conversion in trimethylation.

The density plots of Fig. 3C&D suggest a slight decrease of H3K27ac which is consistent with the western blots shown in Fig. 2B.

If antibodies against EZHIP did not work, I would have strongly appreciated an EZHIP ChIP-seq attempt using a tagged version of EZHIP in EZHIP^{-/-} cells.

As explained in details in the previous point-by-point responses to the reviewers, it is actually likely that the EZHIP ChIP-seq worked, however it shows a broad distribution not sufficiently distinguishable from the input to make definitive conclusion. Besides, our immunofluorescence experiments revealed only a partial overlap between EZHIP and PRC2/H3K27me2 consistent with the hypothesis that EZHIP is not specifically enriched at Polycomb targets.

We did not mention it but we have done the suggested experiment and did not observe any specific enrichment at PRC2 targeted regions.

Importantly, similar debate took place regarding the mechanism of PRC2 inhibition by H3.3-K27M considering that the mutant histone was not found specifically enriched at PRC2 targets.

However, it is now shown that stable colocalization as detected by ChIP is not required for H3.3-mediated PRC2 inhibition (See Stafford et al. 2018 and Piunti et al. 2017).

Reviewer #4 (Remarks to the Author):

Ragazzini et al. present exciting new work characterising a new PRC2 complex member EZHIP. They present quite detailed characterisation in a model cell line and create a knockout mouse model, which has an intriguing phenotype.

In my view (as a new reviewer added at the rebuttal phase), the authors have dealt with the majority of the previous reviewer's comments, and added extra data that has improved the manuscript.

We thank the reviewer for the positive comments on the manuscript and for acknowledging that we added significant data for the comprehensive characterization of *Ezhip* and its loss *in vivo*.

While I agree that a full description of the underlying mechanisms is too high a bar (especially given the novelty of the findings) I do feel that some further basic characterisation should be included:

1. PGC expression: Given that the authors have established a role for EZHIP in regulating H3K27me3 levels in mouse germ cells, I think it would be helpful to include a fuller characterisation of gene expression in PGCs. Is it expressed in newly specified PGCs, during migration or only after genital ridge colonisation? Ideally this could be shown by immunostaining at the relevant stages (with knockout controls), however it may be possible to mine RNA-seq datasets that already exist in the literature (please note I do not think that Supp Figure 1G is sufficient in this regard). Although EZHIP is one of the so-called germline-reprogramming-responsive genes, it is possible the gene is lowly expressed prior to its reprogramming associated activation.

Our mouse antibody generated against EZHIP is not of sufficient specificity for immunostaining (See for instance Fig. S5C showing the recognition of non-specific band in western blot). Hence, even if indeed appropriate, we cannot proceed to a better characterization of the protein pattern during the formation of PGCs or after their entering in the genital ridges. Therefore, we processed to look at *Ezhip* expression in publicly available RNA-seq as suggested by the reviewer. We analyzed two independent sets of data: one from the Hajkova group and one from the Saitou lab (accession numbers: GSE76973 and GSE94136, respectively). This analysis revealed that *Ezhip* is very lowly expressed in migrating PGCs with its expression slowly increasing after they entered the genital ridges (see Rebuttal Figure, and shown in the manuscript, specifically for E9.5 – E11.5 PGCs and female PGCs and germ cells from E12.5 – E15.5, in Figure 6A). Importantly, *Ezhip* expression stays comparatively very low (from 1 to 3 log¹⁰ in fold expression) to *Ezh2* or to markers of germ cells (namely, *Dppa3*, *Ddx4* and *Pou5f1*) during the phase of the well know epigenetic reprogramming (E10.5 to E12.5; Hajkova et al 2008). Its expression increase by E13.5 and mirror the one of *Ezh2* (E14.5 and E15.5), but is still significantly low compared to what can be found in post natal oocytes at different stage of differentiation and growth (non-growing oocytes, growing oocytes, and fully grown oocytes) (see left and right graphs of new panel A in Figure 6, which displays normalized expression data from GSE94136 (Saitou's lab) and GSE70116 (Kelsey's lab). Although this data mining is not sufficient to describe the role of *Ezhip* in early germ cells, these results could suggest that *Ezhip* might rather be involved in the stages of the maturation of the oocyte chromatin landscape (so late in germ cells that are already in meiosis) rather than be part of

the epigenetic reprogramming that PGCs (early germ cells) undergo. We integrated these new analysis in Fig 6A and accordingly change the text of the manuscript.
 Line 340 to 350.

2. H3K27me3 in PGCs: At what developmental stage is the first detectable increase in H3K27me3 observed in EZHIP knockout germ cells? Are there already elevated levels in PGCs? Please note, I'm sure the authors are keen to characterise epigenetic reprogramming in PGCs and in the zygote in these knockouts – and I think it is reasonable to save this for a follow up study. However, I do not think establishing when the epigenetic mark regulated by EZHIP is first affected during PGC/germ cell development, is an unreasonable request. This is especially relevant given the statement in the abstract 'a novel functional player in the comprehensive chromatin remodelling that occurs in the gonad'.

We thank the reviewer in this comment (as for the previous) to point out for a putatively important role of *Ezh1p* concerning chromatin composition during germ cell formation besides its involvement in gametes that we describe. This shows the interest of the scientific community when characterizing a new player in oocyte maturation. The present manuscript already entails the characterization of H3K27me3 dynamic in U2OS cells, in sperm and in postnatal stages of oogenesis. Our work stands for a role of *Ezh1p* in chromatin regulation in gametes, and more

generally for its importance in mouse fertility. We believe that further work for different stages of development might indeed reveal other important functions, such as the involvement of *Ezhip* in chromatin reprogramming in E10.5-E12.5 PGCs, later during meiosis (whether during foetal life or post natal) and during the early embryonic chromatin reprogramming after fertilization, but these questions are beyond the scope of this manuscript. We completely agree that it set up a solid case for future investigations in these epigenetic area.

3. H3K27me3 in 4month KO oocytes: I did not think the request by Reviewer #2 for analysis of H3K27me3 in 4month KO oocytes was wholly unreasonable – and the authors should consider whether this could be done. Perhaps, if my points 1 and 2 are adequately addressed, this could be avoided.

We agree that this is an interesting question as we already developed. It must be underlined that considering the phenotype of subfertility of KO/KO females it does require time to obtain sufficient animals for experimental purposes. We made the choice considering the first round of revision conducted with reviewer 2 (now not available), to focus on the transcription (single-oocyte RNA-seq) and to address chromosomal alignments at meiosis than to address these questions at the time.

With these additions I would consider the paper ready for publication in Nature Communications:

Minor points:

1. Abstract line 45: Evidences?

We assumed that the reviewer questioned whether the reported transcriptomic alterations in a subset of oocytes knockout for EZHIP as well as the increase chromosome abnormalities could be qualified as “evidences” that mature oocytes *Ezhip* *-/-* are not fully functional.

We reformulated the sentence to address his/her concern.

Line 48 to 50

2. Line 114: Mammalian seems a broad statement here. Is this based on the expression shown in human also? It is hard to argue function based on expression only and if a broader statement is to be made about mammals I would include germ cell expression data in a broader range of species.

We reformulated the sentence to address this comment.

Line 116 to 117

3. Line 286-7: Does the deletion remove the conserved amino acid region?

Yes, as indicated in the result section:

“with the notable exception of mutant M5 **that lacks the conserved amino-acids stretch** (Fig. 2C)”

4. Line 325: ‘Spermatozoa motility felt within normal standards’: This does sound like a somewhat subjective measure? Unless perhaps there is a typographical error?

This was indeed a typographical error “felt” instead of “fell”, nonetheless we reformulated the sentence.

Line 323